# Scale-wise Distillation of Diffusion Models

**Nikita Starodubcev**
Yandex Research

**Ilya Drobyshevskiy**
HSE University & Yandex Research

**Denis Kuznedelev**
Yandex Research

**Artem Babenko**
Yandex Research

**Dmitry Baranchuk**
Yandex Research

## Abstract

Recent diffusion distillation methods have achieved remarkable progress, enabling high-quality $\sim$4-step sampling for large-scale text-conditional image and video diffusion models. However, further reducing the number of sampling steps becomes more and more challenging, suggesting that efficiency gains may be better mined along other model axes. Motivated by this perspective, we introduce SwD, a scale-wise diffusion distillation framework that equips few-step models with progressive generation, avoiding redundant computations at intermediate diffusion timesteps. Beyond efficiency, SwD enriches the family of distribution matching distillation approaches by introducing a simple patch-level distillation objective based on Maximum Mean Discrepancy (MMD). This objective significantly improves the convergence of existing distillation methods and performs surprisingly well in isolation, offering a competitive baseline for diffusion distillation. Applied to state-of-the-art text-to-image/video diffusion models, SwD approaches the sampling speed of two full-resolution steps and largely outperforms alternatives under the same compute budget, as evidenced by automatic metrics and human preference studies. Project page: `https://yandex-research.github.io/swd`.

## 1 Introduction

Designing large-scale diffusion models (DMs) for high-resolution image and video generation is inherently challenging due to the slow sequential sampling requiring $20-50$ steps. Although state-of-the-art DMs typically operate in $\sim$8$\times$ lower-resolution VAE latent spaces (Rombach et al., 2021), generation speed remains a significant bottleneck, especially for recent large-scale models with $>$8 billion parameters (Sauer et al., 2024; Black Forest Labs, 2024; Cai et al., 2025; Wu et al., 2025). The challenge becomes several times more pronounced for video DMs (Polyak et al., 2024a; Wan et al., 2025; Kong et al., 2024), as the latent resolution also scales across the temporal dimension.

Previous works have made substantial efforts in DM acceleration from different perspectives (Lu et al., 2022; Song et al., 2020a; Wimbauer et al., 2024; Li et al., 2023; Jin et al., 2025). One of the most successful directions is distilling DMs into few-step generators (Song et al., 2023; Sauer et al., 2023; Yin et al., 2024b; Kim et al., 2024), where recent methods (Yin et al., 2024a; 2025) achieve state-of-the-art performance of large-scale text-conditional DMs for $\sim$4 steps. Notably, these approaches generally focus on reducing the number of sampling steps while freezing other promising degrees of freedom, such as model architectures or input resolution.

Recently, Rissanen et al. (2023); Dieleman (2024) drew parallels between the coarse-to-fine nature of image diffusion and implicit spectral autoregression. Specifically, the reverse diffusion process was shown to progressively predict higher spatial frequencies conditioned on previously generated lower frequencies via the input. This observation connects DMs to next-scale prediction models (Tian et al., 2024; Voronov et al., 2024; Han et al., 2024), which also sample higher spatial frequencies at each step, but do so explicitly via upscaling. Therefore, these models suggest significant efficiency potential by performing most steps at lower resolutions. However, despite this connection, state-of-the-art few-step DMs still operate at a fixed resolution throughout the diffusion process, highlighting an underexplored direction for improving their efficiency.

**Contribution.** Since most state-of-the-art DMs belong to the latent diffusion family (Rombach et al., 2021), we first address whether the spectral autoregression perspective also applies to latent representations. In this work, we conduct a spectral analysis of existing VAE latent spaces and also extend it to the video domain. Our findings confirm that both spatial and temporal latent resolutions implicitly increase over the diffusion process, similarly to the natural images. This suggests that latent DMs can avoid redundant computations at intermediate noisy timesteps, where high frequencies are largely suppressed.

Motivated by this observation, we introduce a *Scale-wise Distillation* (SwD) framework, which transforms an arbitrary pretrained DM into a single few-step model that progressively increases spatial and temporal sample resolutions at each generation step. SwD integrates seamlessly with existing distribution matching distillation approaches (Sauer et al., 2023; Yin et al., 2024b) and leverages their few-step sampling algorithms, which appear naturally aligned with progressive generation.

In addition to the scale-wise distillation framework, we present a simple yet surprisingly effective diffusion distillation objective that minimizes Maximum Mean Discrepancy (MMD) (Gretton et al., 2012) in the feature space of a pretrained DM. The proposed objective complements state-of-the-art distillation methods and achieves strong performance even in isolation, establishing a competitive baseline for DM distillation. Importantly, it requires no additional trainable models, making it computationally efficient and easy to combine with existing distillation pipelines.

We apply SwD to state-of-the-art text-to-image and video DMs and show that our models compete or even outperform their teachers being more than $10\times$ faster. Compared to full-resolution few-step models, SwD significantly surpasses them under a similar computational budget. For the same number of sampling steps, SwD provides $\sim 2\times$ speedup in text-to-image generation and $\sim 3\times$ in text-to-video generation, without compromising quality.

## 2 RELATED WORK

**Diffusion distillation into few-step models.** Diffusion distillation methods aim to reduce generation steps to $1-4$ while maintaining teacher model performance. These methods can be largely grouped into two categories: *teacher-following* methods (Meng et al., 2023; Song et al., 2023; Luo et al., 2023a; Huang et al., 2023; Song & Dhariwal, 2024; Kim et al., 2025) and *distribution matching* (Yin et al., 2024b;a; Sauer et al., 2023; 2024; Luo et al., 2023b; Zhou et al., 2024b;a).

Teacher-following methods approximate the teacher's noise-to-data mapping by integrating the diffusion ODE in fewer steps than numerical solvers (Song et al., 2020a; Lu et al., 2022). Distribution matching methods relax the teacher-following constraint, focusing instead on aligning student and teacher distributions without requiring exact noise-to-data mapping. State-of-the-art approaches, such as DMD2 (Yin et al., 2024a) and ADD (Sauer et al., 2023; 2024), demonstrate strong generative performance in $\sim 4$ steps. However, they still exhibit noticeable quality degradation at $1-2$ step generation, leaving room for further improvement. Recently, DMD has been successfully adopted for video diffusion models (Yin et al., 2025; Huang et al., 2025).

**Progressive generation with DMs.** The idea of progressively increasing resolution during diffusion generation was initially exploited in hierarchical or cascaded DMs (Ho et al., 2021; Saharia et al., 2022; Ramesh et al., 2022; Kastryulin et al., 2025; Gu et al., 2023b), which are strong competitors to latent DMs (Rombach et al., 2021) for high-resolution generation. Cascaded DMs consist of multiple DMs operating at different resolutions, where each model performs a diffusion sampling from scratch, conditioned on the lower-resolution sample. To bridge progressive generation with diffusion processes, several works (Gu et al., 2023a; Teng et al., 2023; Atzmon et al., 2024; Jin et al., 2025; Zhang et al., 2025; Anagnostidis et al., 2025; Haji-Ali et al., 2025) have presented multi-stage pipelines, where DMs are trained to smoothly transition to higher-resolution noisy samples during diffusion sampling. SwD follows up this line of research by proposing a framework that readily integrates into existing diffusion distillation procedures and adapts arbitrary pretrained DMs into progressive few-step models.

**Maximum Mean Discrepancy in generative modeling.** Maximum Mean Discrepancy (MMD) is a metric between two distributions $P$ and $Q$, widely explored in early GAN works (Bińkowski et al., 2018; Wang et al., 2018; Dziugaite et al., 2015; Bellemare et al., 2017; Sutherland et al., 2016).

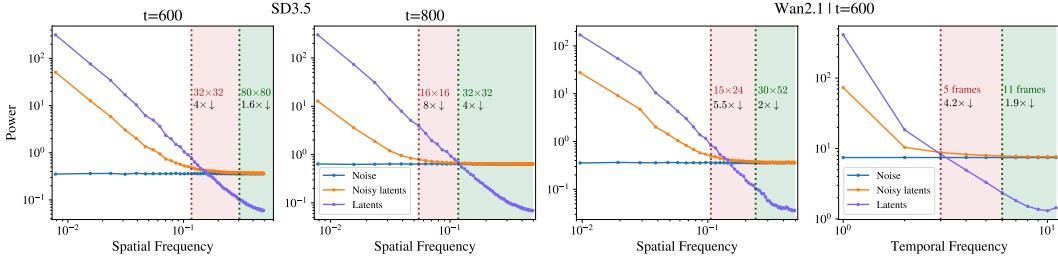

Figure 1: **Spectral analysis** of SD3.5 VAE latents ($128{\times}128$) (*Left*) and Wan2.1 ($21{\times}60{\times}104$) for spatial and temporal dimensions (*Right*). Vertical lines mark the frequency boundaries for which the frequency components to the right are not present in lower resolution latents. Noise masks high frequencies, suggesting that latent DMs can operate at lower latent resolutions for high noise levels. Green area indicates the allowed latent resolution at corresponding timestep, while Red area shows that further resolution reduction would lead to noticeable information loss.

Given a positive-definite *kernel function* $k(\mathbf{x}, \mathbf{y})$, the MMD can be defined as

$$\mathrm{MMD}^2(P, Q) = \mathbb{E}_{\mathbf{x},\mathbf{x}'\sim P}[k(\mathbf{x}, \mathbf{x}')] + \mathbb{E}_{\mathbf{y},\mathbf{y}'\sim Q}[k(\mathbf{y}, \mathbf{y}')] - 2\,\mathbb{E}_{\mathbf{x}\sim P, \mathbf{y}\sim Q}[k(\mathbf{x}, \mathbf{y})], \qquad (1)$$

where $\mathbf{x}$ and $\mathbf{y}$ denote samples from the generated and target distributions, respectively.

Generative Moment Matching Networks (GMMNs) (Li et al., 2015) employ the MMD with a fixed Gaussian kernel (RBF) directly in data space. GANs, in contrast, typically consider learnable kernels, designed as the composition of a discriminator with a fixed kernel.

In diffusion modeling, the MMD has been explored for DM training (Bortoli et al., 2025) or finetuning (Aiello et al., 2023). DMMD (Galashov et al., 2025) employs noise-adapted discriminators for MMD gradient flows (Arbel et al., 2019). Recently, IMM (Zhou et al., 2025) leveraged the MMD for consistency distillation (Song et al., 2023), computing the MMD with a fixed kernel between raw generator predictions at different timesteps. In our work, we adopt the MMD between student and teacher distributions in the feature space of a pretrained DM, yielding a powerful and effective distribution matching objective.

## 3 LATENT SPACE SPECTRAL ANALYSIS

Rissanen et al. (2023) and Dieleman (2024) showed that, in pixel space, diffusion models approximate spectral autoregression for natural images. Since state-of-the-art text-conditional diffusion models operate on VAE latent representations (Rombach et al., 2021), we first investigate this spectral perspective for various latent spaces and also extend it to the video domain.

Following Dieleman (2024), we evaluate *radially averaged power spectral density* (RAPSD), i.e., the averaged spectra power across different spatial frequency components, and its one-dimensional analogue for temporal frequencies in video.

We examine the latent spaces of image and video diffusion models, specifically Stable Diffusion 3.5 (SD3.5) (Esser et al., 2024) and Wan2.1 (Wan et al., 2025). The SD3.5 VAE maps $3{\times}1024{\times}1024$ images into $16{\times}128{\times}128$ latents, while the Wan2.1 VAE encodes $81{\times}3{\times}480{\times}832$ video inputs into $21{\times}16{\times}60{\times}104$ latents. Both models use a flow-matching process (Lipman et al., 2023).

Figure 1 shows the RAPSD of Gaussian noise (blue), clean latents (purple) and noisy latents (orange) at different timesteps. Figure 1 (Left) provides the results for SD3.5 VAE latents. Figure 1 (Right) shows RAPSD across both spatial and temporal frequencies of Wan2.1 latents. Vertical lines indicate frequency boundaries: the components to the right correspond to high frequencies absent at lower resolutions, while those to the left align with the full latent resolution ($128{\times}128$).

Additional results, including more timesteps and SDXL (Podell et al., 2024) latents under a variance-preserving diffusion process (Ho et al., 2020; Song et al., 2020b), are provided in Appendix E.

**Observations.** First, we note that the latent frequency spectrum approximately follows a power law, similar to natural images (van der Arjen Schaaf & van Johannes Hateren, 1996). In contrast, how-

ever, highest frequency components in latent space exhibit slightly greater magnitude. We attribute this to the VAE regularization terms, which may cause "clean" latents to appear slightly noisy.

We also observe that the noising process progressively filters out high frequencies, thereby determining the safe downsampling range without noticeable information loss. Figure 1 (Left) shows that at $t=800$, noise masks high frequency components emerging at resolutions above $32\times32$. This allows for $4\times$ downsampling of $128\times128$ latents (green area). On the other hand, $8\times$ downsampling would corrupt the data signal (red area), as the noise does not fully suppress those frequencies.

A similar effect is observed along the temporal dimension in Wan2.1 latents, see Figure 1 (Right). At $t=600$, the effective signal can be represented with $\sim11$ latent frames instead of the original 21.

**Practical implication.** Based on this analysis, we suppose that latent diffusion models may operate at lower resolution at high noise levels without losing the data signal. In other words, modeling high frequencies at timestep $t$ is unnecessary if those frequencies are already masked at that noise level. Note that this holds true for both spatial and temporal axes for video DMs. We summarize this conclusion as follows:

> Latent diffusion allows lower-resolution modeling at high noise levels across spatial and temporal dimensions.

## 4 METHOD

This section introduces a *scale-wise distillation*, SwD, framework for diffusion models. We begin by describing the SwD pipeline, highlighting its key features and challenges. Then, we present our distillation objective based on Maximum Mean Discrepancy (MMD).

### 4.1 SCALE-WISE DISTILLATION OF DMS

The core design principle of SwD is to unify multi-scale generation within a *single few-step model* and *single diffusion process*, in contrast to cascaded approaches. To this end, we define a few-step *timestep schedule*, $[t_1, \ldots, t_N]$, and pair each diffusion timestep $t_i$ with a latent resolution $s_i$ from a non-decreasing *scale schedule*, $[s_1, \ldots, s_N]$. Therefore, starting the generation with Gaussian noise at the lowest scale, $s_1$, the resolution of intermediate noisy latents $\mathbf{x}_{t_i}$ is progressively increased over sampling steps.

**Upsampling strategy.** We first address *how to upsample intermediate $\mathbf{x}_{t_i}$ to obtain faithful higher-resolution noisy samples?* A naive approach is to directly upsample $\mathbf{x}_{t_i}$. However, this distorts the noise variance and introduces local noise correlations. Consequently, recent progressive DMs (Jin et al., 2025; Atzmon et al., 2024) propose dedicated techniques to handle jump points and preserve sampling continuity.

In contrast, few-step models enable *stochastic multistep sampling* (Song et al., 2023) by predicting a clean sample $\hat{\mathbf{x}}_0$ and then renoising it to the next noise level. This approach naturally suggests upsampling $\hat{\mathbf{x}}_0$ prior to renoising, thereby preserving the correct noise statistics at higher resolution.

| Configuration | $t = 400$ | $t = 600$ | $t = 800$ |
|---|---|---|---|
| **A** $\mathbf{x}_0 \xrightarrow{\text{noise}} \mathbf{x}_t$ | 9.2 | 10.3 | 13.7 |
| **B** $\mathbf{x}_0^{\text{down}} \xrightarrow{\text{upscale}} \mathbf{x}_0 \xrightarrow{\text{noise}} \mathbf{x}_t$ | 30.9 | 18.8 | 14.7 |
| **C** $\mathbf{x}_0^{\text{down}} \xrightarrow{\text{noise}} \mathbf{x}_t^{\text{down}} \xrightarrow{\text{upscale}} \mathbf{x}_t$ | 122 | 223 | 327 |

Table 1: Comparison of noisy latent upsampling strategies (**B**, **C**) for $64\to128$ in generation quality (FID-5K). Upsampling $\mathbf{x}_0^{\text{down}}$ before noise injection (**B**) aligns better with original full-resolution noisy latents (**A**).

To validate this intuition, we generate images with Stable Diffusion 3.5 (Esser et al., 2024) from intermediate noisy latents, $\mathbf{x}_t$, obtained with different upsampling strategies. Specifically, given a full-resolution ($128\times128$) real image latent, $\mathbf{x}_0$, and its downscaled version ($64\times64$), $\mathbf{x}_0^{\text{down}}$, we consider: (**A**) a reference setting where noise is added to the original $\mathbf{x}_0$; (**B**) upsampling $\mathbf{x}_0^{\text{down}}$ followed by noise injection; (**C**) injecting noise into $\mathbf{x}_0^{\text{down}}$ followed by upsampling.

As shown in Table 1, the naive strategy (**C**) produces largely out-of-distribution (OOD) noisy latents. In contrast, strategy (**B**) shows comparable results to the original noisy latents at high noise levels,

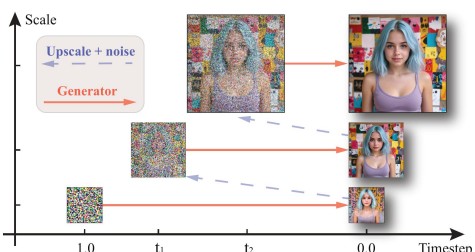

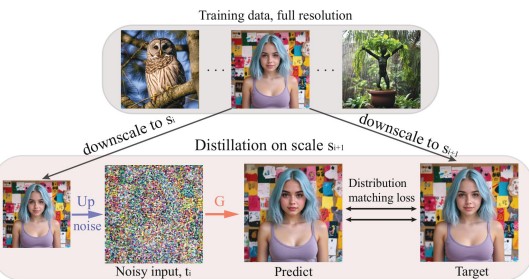

Figure 2: **SwD sampling.** Few-step model starts sampling from noise at the low resolution $s_1$ and gradually increases it over generation steps. At each step, the previous denoised prediction at the scale $s_{i-1}$ is upsampled and noised according to the *timestep schedule*, $t_i$. Then, the generator predicts a clean image at the current resolution $s_i$.

Figure 3: **SwD training step. i)** Sample a pair of adjacent resolutions $[s_i, s_{i+1}]$ from *scale schedule*. **ii)** Downscale the training images to $s_i$ and $s_{i+1}$. **iii)** The lower scale versions are upsampled and noised to a timestep $t_i$ with the forward process. **iv)** Given the noised images, the model $G$ predicts clean data at target scale $s_{i+1}$. **v)** Distribution matching loss is calculated between predicted and target images.

likely because heavy noise masks interpolation artifacts. The results for other upscale ratios are presented in Appendix F.

Overall, we propose to handle scale-transition points by upsampling $\hat{\mathbf{x}}_0$ predictions, followed by renoising. In practice, we use *bicubic interpolation* for spatial dimensions and *adjacent frame blending* for the temporal one.

**Sampling.** SwD performs *progressive stochastic multistep sampling* under the discussed scale-transition approach. Specifically, given the intermediate noisy sample $\hat{\mathbf{x}}_{t_{i-1}}$ at resolution $s_{i-1}$, the model predicts a denoised sample $\hat{\mathbf{x}}_0^{i-1}$. To proceed to the next timestep $t_i$, $\hat{\mathbf{x}}_0^{i-1}$ is upsampled to $s_i$ and renoised with the forward diffusion process, resulting in a higher-resolution sample $\hat{\mathbf{x}}_{t_i}$. Then, the model predicts the next $\hat{\mathbf{x}}_0^i$. Figure 2 illustrates this sampling process.

Although this procedure can be applied directly to pretrained few-step models, we observe in practice that renoising is insufficient to completely remove upsampling artifacts, unless very high noise levels are used. Therefore, we aim to train a few-step generator that also serves as a robust upscaler.

**Training.** We train a single model across multiple resolutions, iterating over pairs of adjacent scales $[s_i, s_{i+1}]$ from the *scale schedule*. At each training step, we sample a batch of full-resolution images or videos, downscale them to the source and target resolutions in pixel space, according to the $s_i$ and $s_{i+1}$ scales, and then encode them into the VAE latent space. Notably, we find that downscaling in pixel space before the VAE encoding largely outperforms latent downscaling in our experiments.

Next, we upsample the lower resolution latents from $s_i$ to $s_{i+1}$ and apply the forward diffusion process according to the *timestep schedule*, $t_i$. The noised latents are then fed into the scale-wise generator, which predicts $\hat{\mathbf{x}}_0$ at the target scale $s_{i+1}$.

Finally, we calculate a distillation loss between the predicted and target latents at $s_{i+1}$. In our work, we use distribution matching, motivated by ADD (Sauer et al., 2023; 2024) and DMD (Yin et al., 2024b;a), achieving state-of-the-art performance in diffusion distillation.

The schematic illustration of this training procedure is provided in Figure 3. Further implementation details and discussions are in Appendix A.

**Discussion on the timestep and scale schedules.** Following Section 3, the emergence of higher-frequency components at lower noise levels can provide useful initial assumptions for designing the schedules. However, since the analysis provides only averaged results and does not account for upsampling artifacts, the schedules ultimately remain hyperparameters.

In practice, we adopt default few-step timestep schedules and slightly shift them toward noisier timesteps, aligning with the intuition that noise injection helps mitigate upsampling artifacts. The scale schedules are progressively increasing, starting with $2-4\times$ lower resolution to achieve the desired inference speedups. We also find that the method is not highly sensitive to specific schedule

choices, allowing them to be selected with minimal tuning and shared across models with similar diffusion processes.

## 4.2 DIFFUSION DISTILLATION WITH MAXIMUM MEAN DISCREPANCY

In addition to the proposed scale-wise distillation framework, we extend the family of distribution matching distillation methods with a patch-level MMD loss, calculated on the intermediate spatial features of the pretrained DMs. Below, we discuss the loss computation for transformer-based DMs (Peebles & Xie, 2022), a dominant architecture in state-of-the-art DMs, and note that its formulation readily extends to convolutional backbones.

First, we leverage the ability of DMs to operate at different noise levels, enabling the extraction of structured and finer-grained signals at high and low noise levels, respectively. Accordingly, before feature extraction, we noise both generated and target samples within a predefined timestep interval. In practice, we find that a low-to-mid noise interval yields slightly better performance.

Then, we extract feature maps $\mathbf{F} \in \mathbb{R}^{N \times L \times C}$ from the middle transformer block of the teacher DM for generated and target samples and denote them as $\mathbf{F}^{\text{fake}}$ and $\mathbf{F}^{\text{real}}$, respectively. $N$ is a batch size, $L$ is a number of spatial tokens, and $C$ is a hidden dimension of the transformer. We then compute the MMD between the distributions of spatial tokens, which correspond to patch representations in vision transformers (Dosovitskiy et al., 2020).

For the MMD computation, we consider two kernels: linear ($k(\mathbf{x}, \mathbf{y}) = \mathbf{x}^T \mathbf{y}$) and the radial basis function (RBF) (Chang et al., 2010). The former aligns feature distribution means, while the latter also matches all higher-order moments. In our experiments, both kernels perform similarly, so we simplify $\mathcal{L}_{\text{MMD}}$ using the linear kernel, i.e., calculate MSE between spatial token means *per image*:

$$\mathcal{L}_{\text{MMD}} = \sum_{n=1}^{N} \left\| \frac{1}{L} \sum_{l=1}^{L} \mathbf{F}_{n,l,\cdot}^{\text{real}} - \frac{1}{L} \sum_{l=1}^{L} \mathbf{F}_{n,l,\cdot}^{\text{fake}} \right\|^2 . \tag{2}$$

Note that the feature means computed across the entire batch rather than per image tend to mitigate condition-specific information, resulting in lower text relevance in our experiments.

**Discussion.** $\mathcal{L}_{\text{MMD}}$ with a linear kernel can be considered as a diffusion distillation adaptation of the feature matching loss, proposed for improved GAN training (Salimans et al., 2016). To our knowledge, such losses have not been explored in the context of diffusion distillation. The notable differences are: i) $\mathcal{L}_{\text{MMD}}$ leverages a pretrained DM instead of a learnable discriminator; ii) it uses the feedback from different noise levels; iii) the feature means are computed per image rather than across the entire batch.

**Overall objective.** We incorporate $\mathcal{L}_{\text{MMD}}$ as an additional loss to $\mathcal{L}_{\text{DMD}}$ and $\mathcal{L}_{\text{GAN}}$ in our scale-wise framework: $\mathcal{L}_{\text{SwD}} = \mathcal{L}_{\text{MMD}} + \alpha \cdot \mathcal{L}_{\text{DMD}} + \beta \cdot \mathcal{L}_{\text{GAN}}$. Interestingly, despite its simplicity, $\mathcal{L}_{\text{MMD}}$ is later shown to be a highly competitive standalone distillation objective.

## 5 EXPERIMENTS

**Models.** We validate our approach in text-to-image generation by distilling SDXL Podell et al. (2024), SD3.5 Medium, SD3.5 Large (Esser et al., 2024) and FLUX.1-dev (Black Forest Labs, 2024). We also apply SwD to the recent text-to-video model, Wan2.1-1.3B (Wan et al., 2025).

**Data.** To remain in the isolated distillation setting and avoid biases from external data, we train all models exclusively on synthetic data generated by their teacher, rather than on real data. We note that this step does not pose a bottleneck for training, as the distillation process itself converges relatively fast ($\sim$3K iterations) and requires significantly less data than the DM training. The synthetic data generation settings for each model are provided in Appendix A.

**Metrics.** For text-to-image models, we use 30K text prompts from the COCO2014 and MJHQ sets (Lin et al., 2015; Li et al., 2024) and evaluate the automatic metrics: FID (Heusel et al., 2017), HPSv3 (Ma et al., 2025), ImageReward (IR) (Xu et al., 2023), and PickScore (PS) (Kirstain et al., 2023) and GenEval (Ghosh et al., 2023). Note that FID was shown to correlate poorly with human perception (Kirstain et al., 2023) for text-to-image assessment but we report it here for completeness.

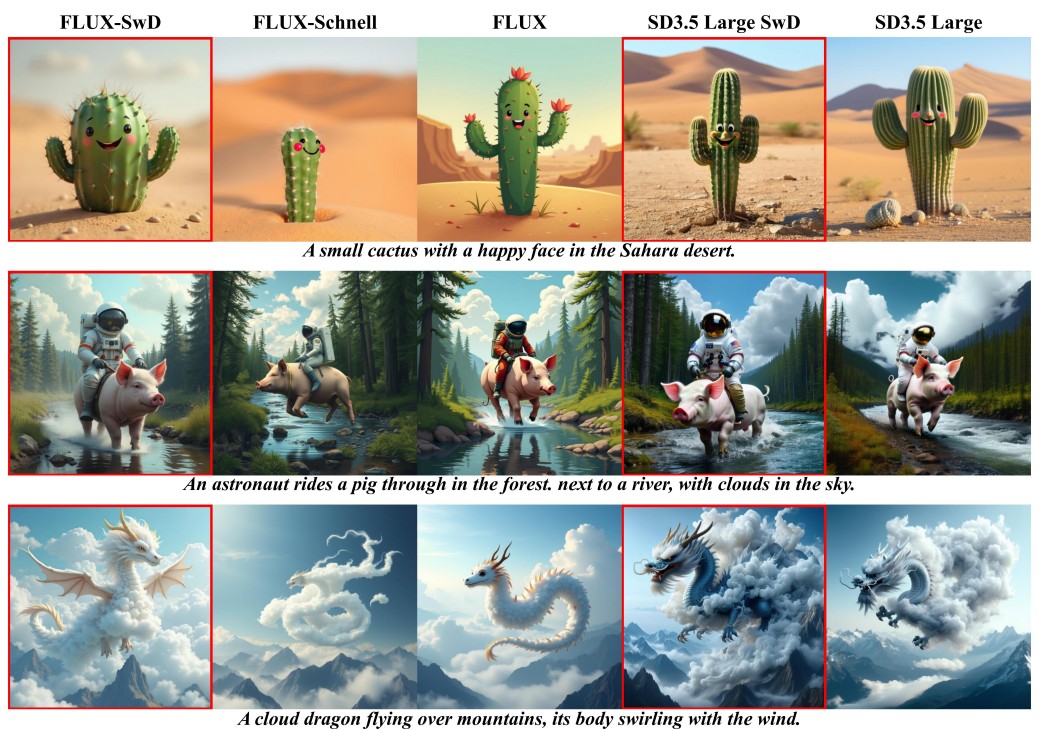

| FLUX-SwD | FLUX-Schnell | FLUX | SD3.5 Large SwD | SD3.5 Large |

*A small cactus with a happy face in the Sahara desert.*

*An astronaut rides a pig through in the forest. next to a river, with clouds in the sky.*

*A cloud dragon flying over mountains, its body swirling with the wind.*

Figure 4: Qualitative results of FLUX-SwD and SD3.5 Large SwD. More examples are in Figure 15.

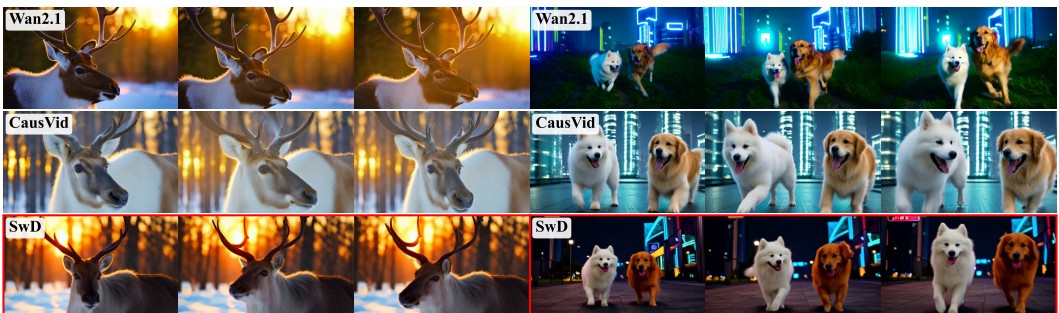

*Cinematic closeup and detailed portrait of a reindeer in a snowy...    A Samoyed and a Golden Retriever dog are playfully romping...*

Figure 5: Qualitative results of Wan2.1-SwD. More examples are in Figure 14.

Also, we conduct a user preference study via side-by-side comparisons evaluated by professional assessors. We select 128 text prompts from the PartiPrompts dataset (Yu et al., 2022), following (Yin et al., 2024a; Sauer et al., 2024), and generate 2 images per prompt. More details are in Appendix H.

For T2V models, we evaluate VBench-2.0 (Zheng et al., 2025), and VisionReward (Xu et al., 2024) and VideoReward (Liu et al., 2025) on 1003 prompts from MovieGenBench (Polyak et al., 2024b).

**Setup.** In our main experiments, we distill the models to 4 or 6 steps. For text-to-image models, the scale schedules begin at image resolutions of $256\times256$ or $512\times512$ and progress to $1024\times1024$. For text-to-video, we start with $21\times160\times272$ and achieve the $81\times480\times832$ resolution. We chose such starting points as lower resolutions provide only marginal speed improvements. The exact timestep and scale schedules for each model are in Appendix C.

Also, we note that SDXL and Wan2.1 are distilled using only the MMD loss, as the DMD loss performs poorly in the scale-wise setting when the base model fails to generate plausible low-resolution images. We discuss this limitation in more detail in Appendix B.

**Baselines.** For text-to-image, we compare with the teacher models and their publicly available distilled versions: DMD2-SDXL (Yin et al., 2024a), SDXL-Turbo (Sauer et al., 2023), Hyper-SD (Ren

| Model | Steps | Latency, s/image | Model size, B | PS ↑ | HPSv3 ↑ | IR ↑ | FID ↓ | PS ↑ | HPSv3 ↑ | IR ↑ | FID ↓ | GenEval ↑ |
|---|---|---|---|---|---|---|---|---|---|---|---|---|
| | | | | COCO 30K | | | | MJHQ 30K | | | | |
| Switti | 14 | 0.44 | 2.5 | 22.6 | 11.1 | 0.98 | 20.0 | 21.6 | 9.8 | 0.84 | 8.9 | 0.62 |
| Infinity | 14 | 0.80 | 2.0 | 22.7 | 11.8 | 0.94 | 28.1 | 21.5 | 10.5 | 0.98 | 12.9 | 0.69 |
| SDXL | 50 | 4.0 | 2.6 | 22.5 | 9.4 | 0.81 | 14.8 | **21.6** | 9.2 | 0.83 | **8.1** | 0.56 |
| SDXL-Turbo | 4 | 0.12* | 2.6 | 22.6 | 10.0 | 0.83 | 17.5 | 21.3 | 9.6 | 0.84 | 15.4 | 0.55 |
| Hyper-SD | 4 | 0.20 | 2.6 | 22.8 | 11.0 | 0.90 | 20.1 | 21.6 | 10.1 | 0.94 | 14.7 | 0.55 |
| SDXL-DMD2 | 4 | 0.20 | 2.6 | 22.8 | 12.0 | 0.87 | 14.1 | 21.6 | 10.1 | 0.86 | 8.3 | **0.58** |
| **SDXL-SwD** | 4 | **0.11** | 2.6 | **22.9** | **12.4** | **0.95** | 21.3 | 21.6 | 10.3 | **0.97** | 15.1 | 0.57 |
| SD3.5-M | 40 | 4.8 | 2.2 | 22.4 | 10.2 | 1.00 | 16.3 | 21.6 | 9.9 | 0.97 | 9.5 | 0.69 |
| SD3.5-M-Turbo | 4 | 0.96 | 2.2 | 22.2 | 9.6 | 0.83 | 17.6 | 21.3 | 9.3 | 0.74 | 13.6 | 0.59 |
| **SD3.5-M-SwD** | 6 | **0.19** | 2.2 | **22.8** | **11.7** | **1.12** | 23.1 | **21.8** | **10.7** | **1.10** | 13.4 | **0.70** |
| SD3.5-L | 28 | 8.3 | 8.0 | **22.8** | 11.3 | 1.06 | 16.5 | 21.8 | 10.4 | 1.04 | 10.7 | 0.70 |
| SD3.5-L-Turbo | 4 | 0.63 | 8.0 | **22.8** | 10 | 0.93 | 22.6 | 21.7 | 9.9 | 0.9 | 13.5 | 0.70 |
| **SD3.5-L-SwD** | 4 | **0.39** | 8.0 | **22.8** | **12.8** | **1.20** | 20.6 | 21.8 | **11.1** | **1.22** | 13.9 | **0.71** |
| FLUX | 30 | 10.0 | 12.0 | 22.9 | 12.4 | 1.03 | 23.6 | 21.7 | 10.7 | 0.93 | 13.0 | 0.66 |
| FLUX-Turbo-Alpha | 8 | 2.75 | 12.0 | **23.1** | 13.4 | 1.08 | 21.2 | 21.5 | 11.2 | 0.97 | 11.3 | 0.66 |
| Hyper-FLUX | 8 | 2.75 | 12.0 | 23.0 | 12.4 | 0.94 | 24.2 | 21.7 | 10.9 | 0.85 | 14.9 | 0.61 |
| FLUX-Schnell | 4 | 1.41 | 12.0 | 22.6 | 11.2 | 1.01 | 16.5 | 21.5 | 10.3 | 0.96 | 9.8 | 0.69 |
| **FLUX-SwD** | 4 | **0.72** | 12.0 | **23.1** | **14.6** | **1.14** | 26.4 | **21.9** | **11.6** | **1.06** | 14.4 | **0.71** |

(*) SDXL-Turbo is the only model that generates at $512{\times}512$, while all other models produce $1024{\times}1024$ images.

Table 3: Quantitative comparison of SwD against other leading open-source models. **Bold** indicates the best-performing model within each DM group, while underline denotes the second best.

et al., 2024), SD3.5-Turbo (Sauer et al., 2024), FLUX-Schnell (Black Forest Labs, 2024), FLUX-Turbo-Alpha (Team, 2024). Also, we evaluate other fast state-of-the-art models, such as next-scale prediction models (Switti (Voronov et al., 2024) and Infinity (Han et al., 2024)).

For the text-to-video task, we compare with the teacher model (Wan2.1-1.3B (Wan et al., 2025)) and its 3-step DMD distilled variant, CausVid (Yin et al., 2025). To ensure a fair comparison, we train CausVid using the official code and setup, but on our training dataset generated with Wan2.1-1.3B, whereas the public version uses high-quality internal data.

## 5.1 MAIN RESULTS

**Text-to-image.** Table 3 and Figure 6 present the comparisons of SwD with the baselines in terms of generation quality and speed. The results are organized into subsections corresponding to different base diffusion models.

We find that SwD models achieve state-of-the-art performance in terms of PS, HPSv3, IR and GenEval within their respective model families. In terms of latency, SwD shows nearly $2\times$ speedup compared to the fastest counterparts.

According to the human study, SwD outperforms most other models in terms of *image complexity* and *image aesthetics*, including the more expensive teachers and their distilled variants, while maintaining comparable levels of *text relevance* and *defects*. We observe a slight degradation in defects for FLUX-SwD compared to the Hyper-FLUX and FLUX models, which are $\sim4\times$ and $\sim14\times$ slower, respectively. SDXL-SwD also has slightly more defects than SDXL-DMD2, which we attribute to the use of the MMD loss alone, as discussed in Section 5.3. Qualitative comparisons are presented in Figure 4, with additional results in Figures 15, 18, and 19.

| Model | Latency, s/video | Vision Reward ↑ | Video Reward ↑ | VBench2 Overall ↑ |
|---|---|---|---|---|
| Wan 2.1 | 137 | 0.038 | 5.43 | 51.6 |
| CausVid* | 4.2 | 0.042 | **6.21** | 52.3 |
| **Spatial SwD** | 2.1 | **0.064** | 6.15 | 52.8 |
| **SwD** | **1.8** | **0.064** | **6.27** | **53.2** |

(*) The version trained on our Wan 2.1 generated dataset using the official implementation and setup.

Table 2: Comparison of 4-step SwD variants with the Wan 2.1 and 3-step CausVid models.

**Text-to-video.** The results in Table 2 show that SwD achieves better performance than the teacher model, while being $72\times$ faster. Compared to CausVid, SwD provides comparable results and achieves $\sim2.3\times$ faster inference. Also, we find that SwD, when applied across both temporal and spatial dimensions, does not degrade quality compared to the spatial-only variant and further improves inference speed. Visual examples are provided in Figures 5 and 14.

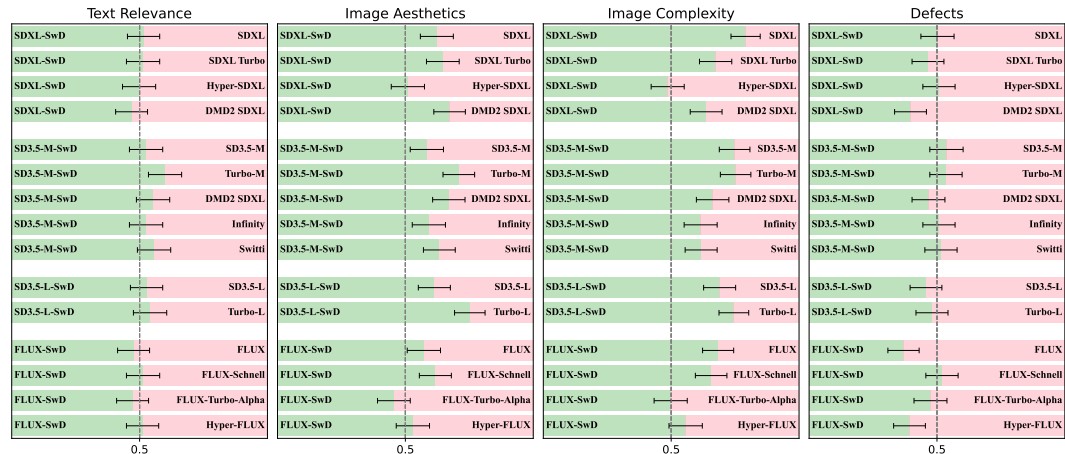

Figure 6: Human preference study for SwD against the baseline models.

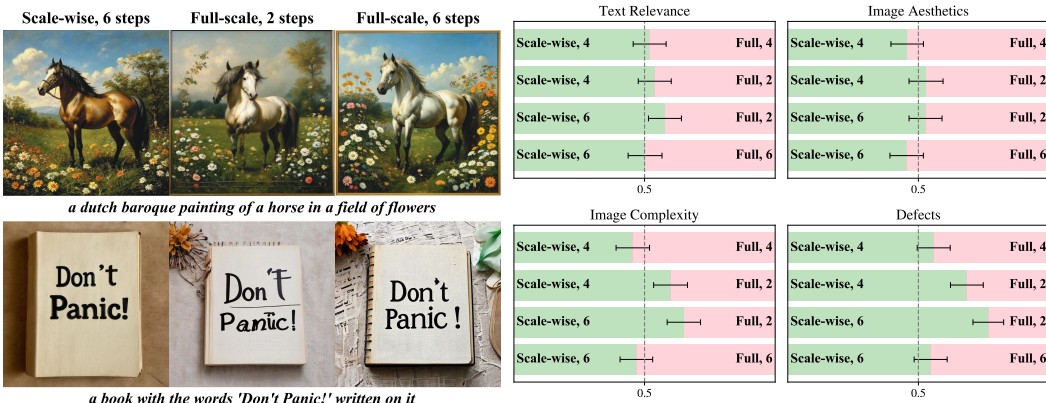

Figure 7: Visual examples (Left) and human preference study (Right) of the scale-wise and full-resolution settings within SD3.5 Medium. The numbers indicate the sampling steps.

## 5.2 SCALE-WISE VERSUS FULL-RESOLUTION

Next, we compare SwD against their full-resolution counterparts. The full-scale baselines use the same timestep schedules but operate at a fixed target latent resolution.

We provide the quality comparisons for the SD3.5 Medium and evaluate FLUX in Appendix D. Comparing the settings for the same number of steps (4 vs 4, 6 vs 6), human evaluation (Figure 7, Right) does not reveal any noticeable quality degradation. Qualitative examples (Figure 7, Left) further confirm this. Interestingly, automatic metrics (Tables 7 and 8) indicate that the scale-wise variants can even outperform their full-resolution counterparts, while being more efficient.

Then, we align generation times of scale-wise and full-resolution setups (4 vs. 2, 6 vs. 2 steps) to assess quality differences. Human evaluation reveals a clear advantage for the scale-wise setup, particularly in reducing *defects* and improving *image complexity*. Examples in Figure 7 (Left) highlight the high defect rates of the 2-step full-resolution baseline. Consistently, automatic metrics also show notable gains in HPSv3 and PS.

**Runtime.** Table 4 reports per-image generation latency (including VAE decoding and text encoding), and Table 5 shows average training iteration time. Compared to the full-resolution setting with the same number of steps (4 steps), the scale-wise setup achieves $\sim 2\times$ speedup in both training and sampling across text-to-image models, and $\sim 3\times$ for text-to-video.

| Setup | Steps | SD3.5-M | SD3.5-L | FLUX | Wan2.1 |
|---|---|---|---|---|---|
| Full-scale | 4 | 0.29 | 0.63 | 1.41 | 5.51 |
| Full-scale | 2 | 0.16 | 0.33 | 0.72 | 2.97 |
| Scale-wise | 6 | 0.19 | 0.41 | 0.97 | 2.61 |
| Scale-wise | 4 | 0.17 | 0.32 | 0.72 | 1.84 |

Table 4: Sampling times (sec / image) of scale-wise and full-resolution setups. The measurement setting is described in Appendix G.

| Setup | Loss | SD3.5-M | SD3.5-L | FLUX | Wan2.1 |
|---|---|---|---|---|---|
| Full-scale | $\mathcal{L}_{\text{SwD}}$ | 7.5 | 13.4 | 22.8 | 70.6 |
| Full-scale | $\mathcal{L}_{\text{SwD-MMD}}$ | 1.0 | 1.7 | 2.9 | 12.7 |
| Scale-wise | $\mathcal{L}_{\text{SwD}}$ | 3.2 | 7.8 | 11.3 | 23.9 |
| Scale-wise | $\mathcal{L}_{\text{SwD-MMD}}$ | 0.4 | 0.9 | 1.4 | 4.4 |

Table 5: Training times (sec / iteration) for scale-wise and full-resolution $4$-step setups using the full objective ($\mathcal{L}_{\text{SwD}}$) and MMD only ($\mathcal{L}_{\text{SwD-MMD}}$).

| $\mathcal{L}_{\text{SwD}}$ setup | PS ↑ | HPSv3 ↑ | IR ↑ | FID ↓ |
|---|---|---|---|---|
| $\mathcal{L}_{\text{SwD}}$ (Main) | **21.8** | **10.7** | 1.11 | **13.6** |
| $\mathcal{L}_{\text{MMD}}$ only | 21.5 | 10.5 | **1.15** | 13.8 |
| $\mathcal{L}_{\text{SwD}}$ w/o $\mathcal{L}_{\text{MMD}}$ | 21.2 | 9.7 | 0.91 | 19.5 |
| $\mathcal{L}_{\text{MMD}}$ Ablation | | | | |
| **A**: RBF kernel | **21.8** | **10.8** | 1.09 | 13.7 |
| **B**: Batch averaging | 21.5 | 10.5 | 0.97 | 16.4 |
| **C**: w/o noising | 21.3 | 10.2 | 1.01 | 16.6 |

Table 6: Ablation study of the $\mathcal{L}_{\text{MMD}}$ objective for SD3.5-Medium SwD on MJHQ30K.

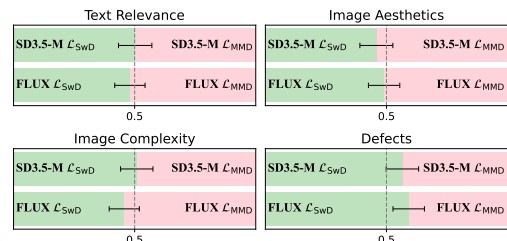

Figure 8: Comparison of the main SwD models against the ones distilled with $\mathcal{L}_{\text{MMD}}$ alone.

## 5.3 Ablation study of MMD loss

Here, we study the role of the $\mathcal{L}_{\text{MMD}}$ loss and its design choices. Most experiments are conducted with the $6$-step SD3.5-M setup from Section 5.1, with the MJHQ results reported in Table 6.

We first assess the $\mathcal{L}_{\text{MMD}}$ contribution to $\mathcal{L}_{\text{SwD}}$. We observe that training with $\mathcal{L}_{\text{MMD}}$ alone slightly underperforms the full $\mathcal{L}_{\text{SwD}}$ but remains effective as an independent distillation method, whereas removing it from $\mathcal{L}_{\text{SwD}}$ leads to a significant drop in performance. Human evaluation (Figure 8) show that $\mathcal{L}_{\text{MMD}}$-only models exhibit noticeable degradation in defects, though not severe. Visual comparisons (Figure 16) confirm that they provide comparable performance to the full $\mathcal{L}_{\text{SwD}}$. Moreover, as shown in Table 5, $\mathcal{L}_{\text{MMD}}$-only training enables more than $7\times$ faster iterations since it avoids training extra models.

Finally, we examine several $\mathcal{L}_{\text{MMD}}$ variants. The $\mathcal{L}_{\text{MMD}}$ with the RBF kernel (**A**) shows similar results. Referring to the feature matching (Salimans et al., 2016), we consider two changes, **B**: the feature tokens in Equation (2) are averaged across the entire batch instead of per image, and **C**: extracting DM features only from clean samples, rather than noising them with the diffusion process. We observe that both **B** and **C** make $\mathcal{L}_{\text{MMD}}$ less effective.

## 6 Conclusion

We introduced SwD, a scale-wise diffusion distillation framework equipped with a novel patch-level MMD-based distillation technique. We show that both components can be readily combined with existing state-of-the-art distillation methods and lead to further efficiency and quality improvements for few-step models. We believe the proposed loss for DM distillation offers substantial potential for further development to pave the way toward a highly effective, self-contained distillation pipeline that eliminates the need for additional trainable models.

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

## A    IMPLEMENTATION DETAILS

Our few-step generator is initialized from a pretrained teacher DM, with trainable LoRA adapters (Hu et al., 2022) added to the attention and MLP layers. We use a LoRA rank of $64$ for SDXL, SD3.5-M, and SD3.5-L, and $128$ for FLUX and Wan 2.1. However, the rank has negligible impact on performance in our experiments.

**Training**. The model is trained to minimize $\mathcal{L}_{\text{SwD}} = \mathcal{L}_{\text{MMD}} + \alpha \cdot \mathcal{L}_{\text{DMD}} + \beta \cdot \mathcal{L}_{\text{GAN}}$.

$\mathcal{L}_{\textbf{DMD}}$ corresponds to the *reverse KL divergence*, which employs the *scores* of the "real" and "fake" distributions. The real score is estimated with the pretrained DM, while the fake score is modeled by training a separate "fake" DM on student-generated samples during distillation. The fake model is parameterized with another set of LoRA adapters added to the teacher DM. Real score estimation uses CFG (Ho & Salimans, 2022) with corresponding default guidance scales for each model.

$\mathcal{L}_{\textbf{GAN}}$. Following DMD2 (Yin et al., 2024a), we also include a GAN loss for text-to-image settings. The discriminator is a $4$-layer MLP head operating on averaged spatial features extracted from the $11$-th (SD3.5-M), $20$-th (SD3.5-L), $15$-th (FLUX, Wan2.1) transformer blocks of the fake DM. For SDXL, the feature maps are extracted from the middle UNet block. The LoRA adapters of the fake DM are also updated using the discriminator loss.

$\mathcal{L}_{\textbf{MMD}}$. For the proposed MMD loss, we use the timestep interval $[0, 600]$ ($[0, 400]$ for SDXL) to noise input samples prior to the feature extraction. The transformer blocks for feature extraction are the same as those used in the GAN setting.

For the SD3.5-M, SD3.5-L, FLUX models, we use all losses in $\mathcal{L}_{\text{SwD}}$. For SDXL and Wan2.1, we use $\mathcal{L}_{\text{MMD}}$ only since $\mathcal{L}_{\text{DMD}}$ degrades the performance for these models as discussed in the following section, while $\mathcal{L}_{\text{GAN}}$ does not show noticeable improvements in our experiments.

Overall, the models are trained with a learning rate of $4e-6$ and batch sizes of $64$ (SD3.5-M, SDXL) and $24$ (FLUX, SD3.5-L, Wan 2.1) for $\sim 3K$ iterations on a single node with 8 A100 GPUs.

**Data.** To exclude the effect of external data, we train the models on the teacher synthetic data, generated prior to distillation using the standard teacher setups. SDXL: 50 sampling steps with a guidance scale of 7.5, followed by 10 refiner steps. SD3.5 Medium: 40 sampling steps with a guidance scale of 4.5. SD3.5 Large: 28 sampling steps with a guidance scale of 4.5. FLUX: 30 sampling steps with a guidance scale of 4.5. Wan2.1: 50 steps with a guidance scale of 5.0.

## B    IMPORTANCE OF SCALE-ADAPTED TEACHER MODELS

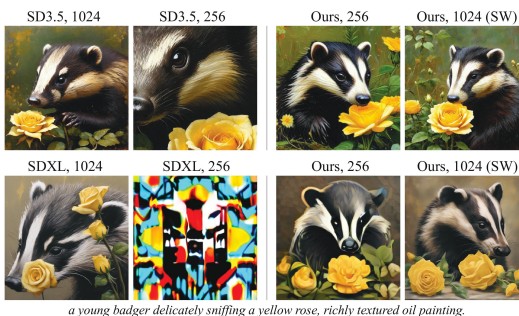

*a young badger delicately sniffing a yellow rose, richly textured oil painting.*

Figure 9: SD3.5 generates cropped images at low-resolutions ($256 \times 256$), while SDXL does not produce meaningful images at all. SwD successfully adapts the model to both fixed low-resolution sampling (Ours, 256) and scale-wise sampling (Ours, 1024 SW) thanks to $\mathcal{L}_{\text{GAN}}$ and $\mathcal{L}_{\text{MMD}}$. In contrast, $\mathcal{L}_{\text{DMD}}$ tends to inherit teacher's limitations and thereby should be applied with caution or disabled altogether in such cases.

We also address the following important question: *Does the teacher model need to be capable of generating images at low scales prior to scale-wise distillation?* The teacher model may not inherently handle low scales effectively, making scale-wise distillation more challenging than the full-scale distillation. If this limitation is significant, additional pretraining of the teacher on small scales might be required.

We evaluate the ability of SD3.5 Medium and SDXL to generate images at a lower resolution ($256 \times 256$). The results are shown in Figure 9.

We observe that SD3.5 produces cropped and simplified images, but the overall quality remains acceptable, likely due to its pretraining at $256 \times 256$ resolution. SwD successfully distills this model, mitigating the cropping problem and increasing image complexity in both the fixed low-resolution setting (Ours, 256) and under scale-wise sampling (Ours, 1024 SW).

| Setup | Steps | PS ↑ | HPSv3 ↑ | IR ↑ | FID ↓ | Setup | Steps | PS ↑ | HPSv3 ↑ | IR ↑ | FID ↓ |
|---|---|---|---|---|---|---|---|---|---|---|---|
| | | | SD3.5 Medium | | | | | | SD3.5 Medium | | |
| Scale-wise | 6 | **22.8** | **11.7** | 1.10 | 23.1 | Scale-wise | 6 | **21.8** | **10.7** | 1.10 | 13.4 |
| Scale-wise | 4 | 22.7 | **11.7** | **1.12** | 23.7 | Scale-wise | 4 | **21.8** | **10.7** | **1.13** | 13.7 |
| Scale-wise | 2 | 22.6 | 10.6 | 1.09 | 22.3 | Scale-wise | 2 | 21.7 | 10.3 | 1.10 | **12.8** |
| Full-scale | 6 | 22.5 | 11.2 | 1.08 | 20.4 | Full-scale | 6 | 21.6 | 10.3 | 1.09 | 13.4 |
| Full-scale | 4 | 22.5 | 11.3 | 1.09 | 21.2 | Full-scale | 4 | 21.7 | 10.4 | 1.10 | 13.5 |
| Full-scale | 2 | 22.3 | 10.8 | 1.03 | **20.3** | Full-scale | 2 | 21.5 | 10.0 | 1.04 | 13.1 |
| | | | FLUX | | | | | | FLUX | | |
| Scale-wise | 4 | **23.1** | **14.6** | **1.14** | **26.4** | Scale-wise | 4 | **21.9** | **11.6** | 1.06 | 14.4 |
| Scale-wise | 2 | 23.0 | 14.1 | 1.12 | 26.5 | Scale-wise | 2 | **21.9** | 11.5 | **1.10** | 14.0 |
| Full-scale | 4 | **23.1** | 14.0 | 1.13 | 28.5 | Full-scale | 4 | 21.8 | 11.3 | 1.09 | 14.4 |
| Full-scale | 2 | 23.0 | 13.8 | 1.13 | 26.9 | Full-scale | 2 | 21.8 | 11.2 | 1.08 | **13.4** |

Table 7: Quantitative comparison between scale-wise and full-scale setups in terms of automatic metrics on COCO30K.

Table 8: Quantitative comparison between scale-wise and full-scale setups in terms of automatic metrics on MJHQ30K.

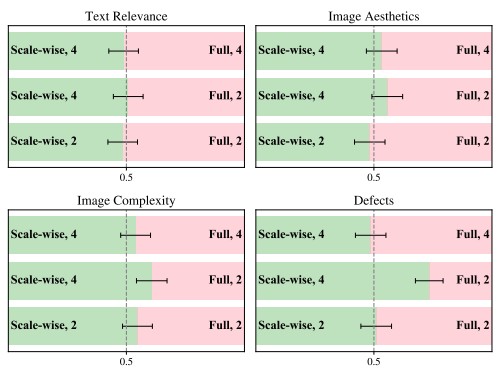

Figure 10: Human preference study comparing scale-wise and full-resolution FLUX setups.

| Setup | PS ↑ | HPSv3 ↑ | IR ↑ | FID ↓ |
|---|---|---|---|---|
| | | COCO2014 | | |
| Main $s=[32, 48, 64, 80, 96, 128]$ | **22.8** | **11.7** | **1.10** | **23.1** |
| $s=[64, 64, 64, 64, 64, 128]$ | 22.4 | 10.3 | 1.02 | 23.7 |
| $s=[32, 32, 32, 32, 32, 128]$ | 22.3 | 9.8 | 0.97 | 23.8 |
| | | MJHQ | | |
| Main $s=[32, 48, 64, 80, 96, 128]$ | **21.8** | **10.7** | **1.11** | **13.6** |
| $s=[64, 64, 64, 64, 64, 128]$ | 21.3 | 9.8 | 1.06 | 14.6 |
| $s=[32, 32, 32, 32, 32, 128]$ | 21.2 | 9.4 | 0.99 | 15.7 |

Table 9: Comparisons to the "constant" scale schedules for SD3.5-Medium SwD.

More importantly, SDXL fails to generate faithful images at $256 \times 256$ resolution. Interestingly, SwD is able to recover from such a poor starting point during distillation.

We attribute SwD 's ability to adapt to lower-resolution sampling to the $\mathcal{L}_{GAN}$ and $\mathcal{L}_{MMD}$ objectives that use low-resolution target images during distillation. In contrast, $\mathcal{L}_{DMD}$ does not rely on reference images at the target resolution. Empirically, we find that $\mathcal{L}_{DMD}$ degrades student performance when applied to teachers that struggle with low-resolution generation (e.g., SDXL and Wan2.1).

In summary, **$\mathcal{L}_{DMD}$ is sensitive to the teacher's ability to generate in low resolution and tends to inherit its limitations, whereas $\mathcal{L}_{GAN}$ and $\mathcal{L}_{MMD}$ are significantly more robust.**

Therefore, in our main experiments, we perform scale-wise distillation for the SDXL and Wan 2.1 models using $\mathcal{L}_{MMD}$ only. We excluded $\mathcal{L}_{GAN}$ for these models as it did not show significant additional improvements.

## C SwD TIMESTEP AND SCALE SCHEDULES

Below, we provide the timestep $t$ and scale $s$ schedules used in our main experiments. The scale schedule shows the resolutions in the corresponding VAE latent spaces.

**SDXL.** $t = [1000, 800, 600, 400]$, $s = [64, 80, 96, 128]$.

**SD3.5 Medium.** $t = [1000, 945, 896, 790, 737, 602]$, $s = [32, 48, 64, 80, 96, 128]$.

**SD3.5 Large.** $t = [1000, 896, 737, 602]$, $s = [64, 80, 96, 128]$.

**FLUX.** $t = [1000, 945, 790, 602]$, $s = [32, 64, 96, 128]$.

**Wan2.1.** $t = [1000, 896, 737, 602]$, $s = [6 \times 20 \times 34, 11 \times 30 \times 52, 16 \times 40 \times 70, 21 \times 60 \times 104]$.

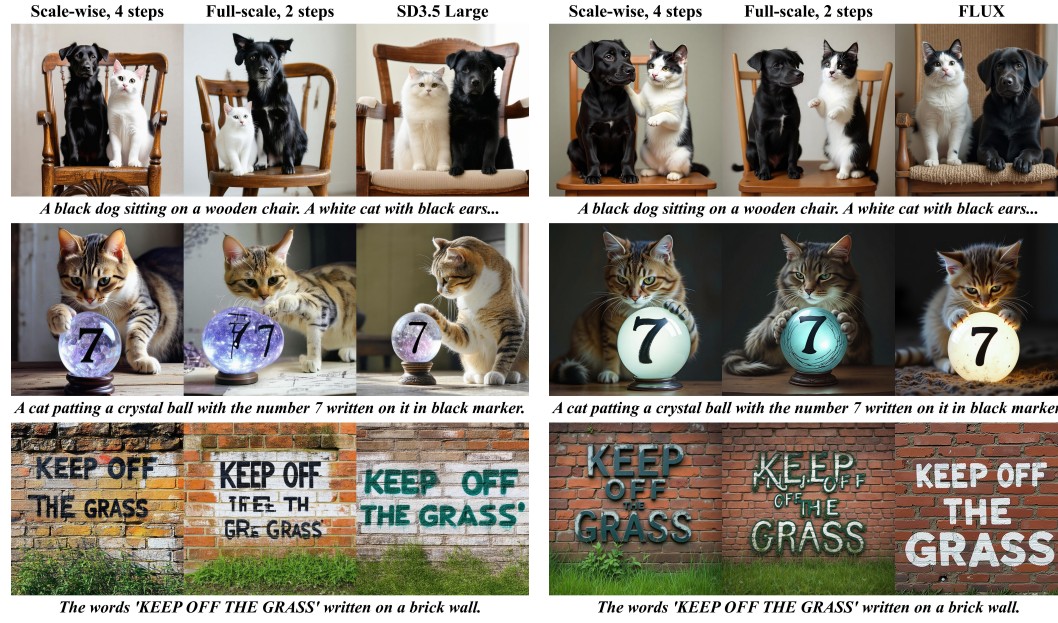

Figure 11: Visual examples of 4-step scale-wise and 2-step full-resolution SD3.5-Large settings.

Figure 12: Visual examples of 4-step scale-wise and 2-step full-resolution FLUX settings.

## D  SCALE-WISE VERSUS FULL-RESOLUTION

Here, we provide additional results to compare the various scale-wise and full-resolution settings. Table 7 and Table 8 present the automatic metrics for SD3.5 Medium and FLUX on the COCO and MJHQ datasets. The visual examples for FLUX, SD3.5 Large, SD3.5 Medium are presented in Figures 11, 12, and 17, respectively.

In Table 9, we evaluate the use of "constant" 6-step scale schedules $s=[64, 64, 64, 64, 64, 128]$ and $s=[32, 32, 32, 32, 32, 128]$ for SD3.5 Medium in contrast to the progressively growing schedule $s=[32, 48, 64, 80, 96, 128]$, used in our main setup. Note that the last step is required to be made at the target resolution. The results show that it is important to gradually increase the resolution over sampling steps.

|  |  | Patch FID ↓ |  | Patch CLIP-IQA ↑ |  |
|---|---|---|---|---|---|
| Model | Steps | SwD | Full-scale | SwD | Full-scale |
| SD3.5-M | 6 | 20.1 | 21.1 | 0.85 | 0.85 |
| SD3.5-L | 4 | 20.4 | 21.9 | 0.88 | 0.88 |
| FLUX | 4 | 18.7 | 18.9 | 0.80 | 0.79 |

Table 10: Comparison of SwD and full-resolution distilled models in high-frequency detail preservation on MJHQ30K.

Finally, we investigate if scale-wise distillation affects high-frequency details compared to the models distilled at the target resolution. We evaluate Patch FID (Lin et al., 2024) and Patch CLIP-IQA (Wang et al., 2023) on MJHQ30K. As shown in Table 10, SwD achieves similar scores to the full-resolution distilled models, indicating no noticeable degradation of high-frequency details.

## E  EXTENDED LATENT SPACE SPECTRAL ANALYSIS

We provide additional results for our latent space spectral analysis in Section 3. Figure 13 provides more results for the SD3.5 and Wan2.1 models and also includes the analysis for FLUX (Black Forest Labs, 2024) and SDXL (Podell et al., 2024). In contrast to other models, the SDXL model (Podell et al., 2024) uses a variance-preserving (VP) diffusion process (Ho et al., 2020; Song et al., 2020b). The SDXL latent space has $C=4$ channels and $128{\times}128$ spatial resolution.

## F  EXTENDED ANALYSIS OF NOISY LATENT UPSCALING STRATEGIES

| Configuration | $t = 400$ | $t = 600$ | $t = 800$ |
|---|---|---|---|
| $\mathbf{x}_0^{128 \times 128} \xrightarrow{\text{noise}} \mathbf{x}_t^{128 \times 128}$ | 9.2 | 10.3 | 13.7 |
| $\mathbf{x}_0^{32 \times 32} \xrightarrow{\text{upscale}} \mathbf{x}_0^{128 \times 128} \xrightarrow{\text{noise}} \mathbf{x}_t^{128 \times 128}$ | 90.2 | 40.8 | 16.6 |
| $\mathbf{x}_0^{32 \times 32} \xrightarrow{\text{noise}} \mathbf{x}_t^{32 \times 32} \xrightarrow{\text{upscale}} \mathbf{x}_t^{128 \times 128}$ | 217 | 298 | 361 |
| $\mathbf{x}_0^{64 \times 64} \xrightarrow{\text{upscale}} \mathbf{x}_0^{128 \times 128} \to \mathbf{x}_t^{128 \times 128}$ | 30.9 | 18.8 | 14.7 |
| $\mathbf{x}_0^{64 \times 64} \xrightarrow{\text{noise}} \mathbf{x}_t^{64 \times 64} \xrightarrow{\text{upscale}} \mathbf{x}_t^{128 \times 128}$ | 122 | 223 | 327 |
| $\mathbf{x}_0^{96 \times 96} \xrightarrow{\text{upscale}} \mathbf{x}_0^{128 \times 128} \xrightarrow{\text{noise}} \mathbf{x}_t^{128 \times 128}$ | 18.8 | 14.3 | 14.3 |
| $\mathbf{x}_0^{96 \times 96} \xrightarrow{\text{noise}} \mathbf{x}_t^{96 \times 96} \xrightarrow{\text{upscale}} \mathbf{x}_t^{128 \times 128}$ | 32.8 | 58.9 | 120 |

Table 11: Extended results of the noisy latent upscaling strategies. We evaluate FID-5K on COCO2014 by generating images with SD3.5-M using $\mathbf{x}_t$ obtained with different upsampling strategies discussed in Section 4.1.

In Table 11, we present the extended comparison of noisy latent upscaling strategies for $32 \times 32 \to 128 \times 128$ and $96 \times 96 \to 128 \times 128$ setups. Specifically, given a full-resolution ($128 \times 128$) real image latent, $\mathbf{x}_0$, and its downscaled versions $\mathbf{x}_0^{32 \times 32}$, $\mathbf{x}_0^{64 \times 64}$ and $\mathbf{x}_0^{128 \times 128}$. The images are generated with Stable Diffusion 3.5 Medium (Esser et al., 2024) from intermediate noisy latents, $\mathbf{x}_t$, obtained with different upscaling strategies. We use a default guidance scale of 7 and the standard 28-step timestep schedule.

Upscaling the noised low-resoluton latent provides poor performance across all upscaling factors. For upscaling $\mathbf{x}_0$-predictions, as expected, lower upscale ratios consistently lead to better results across all timesteps, since the predictions are less distorted by upscaling. Also, we note that even for the $96 \times 96$ setup, the performance is still not fully recovered at $t=600$, suggesting that the upsampled latents are still slightly shifted and require the scale-wise model training.

## G  RUNTIME MEASUREMENT SETUP

In our experiments, we measure runtimes in half-precision (FP16), using *torch.compile* for all models: VAE decoders, text encoders, and generators. Note that, under our very fast sampling settings, the computational costs of the text encoder and VAE start to account for a noticeable portion of the overall runtime. Thus, we replace original VAEs with TinyVAEs (Boer Bohan, 2025) for all models.

The measurements are conducted in an isolated environment on a single A100 GPU. We use a batch size of 8 for all runs, and each measurement is averaged over 100 independent runs. The latency is then obtained by dividing the average runtime by the batch size.

## H  HUMAN EVALUATION

The evaluation is conducted using Side-by-Side (SbS) comparisons, where assessors are presented with two images alongside a textual prompt and asked to choose the preferred one. For each pair, three independent responses are collected, and the final decision is determined through majority voting.

The human evaluation is carried out by professional assessors who are formally hired, compensated with competitive salaries, and fully informed about potential risks. Each assessor undergoes detailed training and testing, including fine-grained instructions for every evaluation aspect, before participating in the main tasks.

In our human preference study, we compare the models across four key criteria: relevance to the textual prompt, presence of defects, image aesthetics, and image complexity. Figures 20, 23, 21, 22 illustrate the interface used for each criterion. Note that the images displayed in the figures are randomly selected for demonstration purposes.

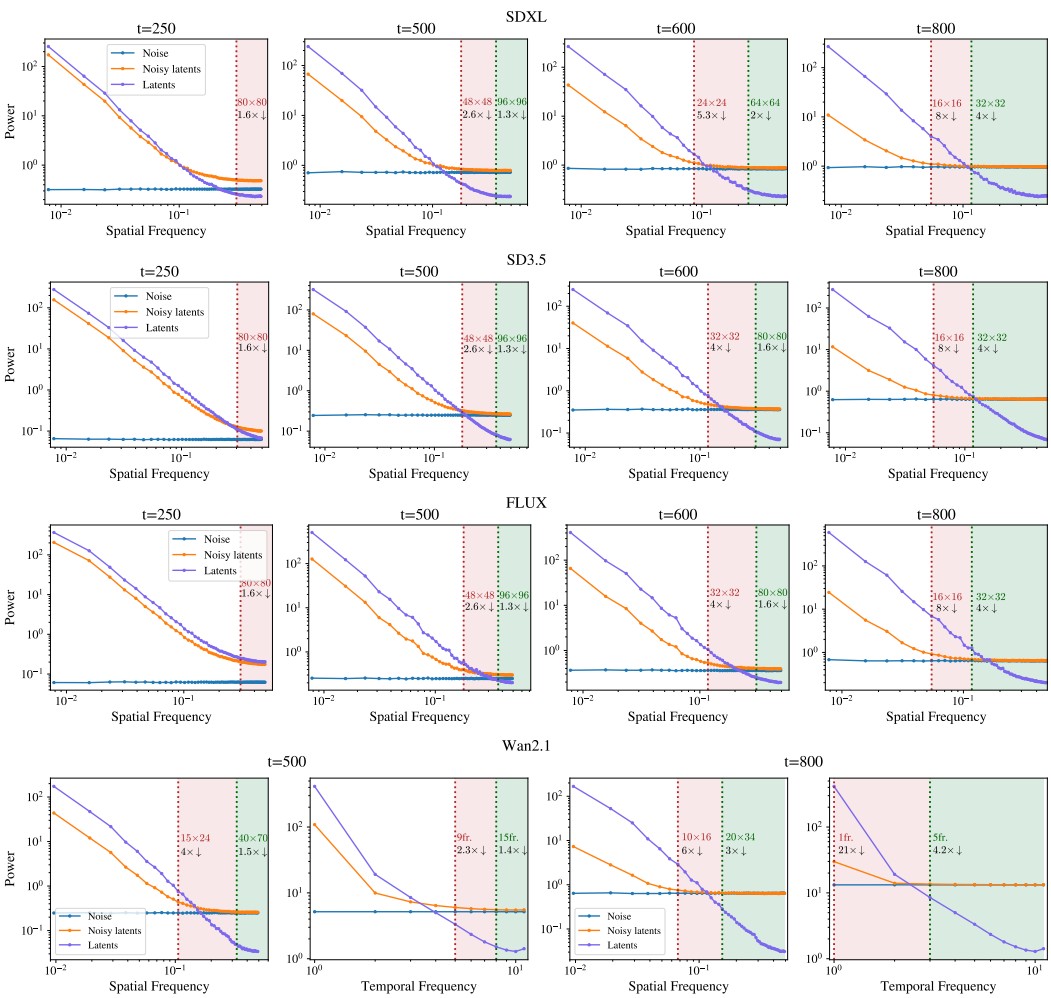

Figure 13: Extended spectral analysis from Section 3 to more timesteps and models (FLUX, SDXL).

| Method | Total Score | Creativity Score | Commonsense Score | Controllability Score | Human Fidelity Score | Physics Score | Human Anatomy | Human Clothes | Human Identity | Composition | Diversity | Mechanics | Material | Thermotics | Multi-View Consistency | Dynamic Spatial Relationship | Dynamic Attribute | Motion Order Understanding | Human Interaction | Complex Landscape | Complex Plot | Camera Motion | Motion Rationality | Instance Preservation |
|---|---|---|---|---|---|---|---|---|---|---|---|---|---|---|---|---|---|---|---|---|---|---|---|---|
| Wan 2.1 | 51.59 | **53.75** | 57.06 | 22.65 | 83.03 | 41.45 | 87.00 | 91.24 | 70.85 | 42.56 | **64.94** | **59.16** | 36.58 | 57.89 | 12.15 | 26.08 | 15.01 | 21.21 | 46.33 | **19.77** | **11.02** | 19.13 | 28.16 | 85.96 |
| CausVid | 52.31 | 39.94 | 59.69 | 21.24 | **92.54** | **48.16** | **91.51** | 97.73 | **88.40** | 46.47 | 33.41 | 53.15 | 37.33 | 63.23 | **38.95** | 26.57 | 17.58 | 18.51 | 53.66 | 18.22 | 7.75 | 6.41 | 27.58 | **91.81** |
| **Spatial SwD** | 52.43 | 44.54 | 60.84 | **29.11** | 84.10 | 43.57 | 84.52 | 94.54 | 73.26 | **53.05** | 36.03 | 58.33 | 44.68 | 56.83 | 14.45 | **35.74** | **27.83** | **21.88** | **71.66** | 19.11 | 10.59 | 16.97 | 29.88 | **91.81** |
| SwD | **53.22** | 40.44 | **63.72** | 26.80 | 88.42 | 46.73 | 89.00 | **99.54** | 76.73 | 47.30 | 33.59 | 57.14 | **47.36** | **63.5** | 18.94 | 33.33 | 17.58 | 20.20 | 71.00 | 14.44 | 10.07 | **20.98** | **35.63** | **91.81** |

Table 12: Full comparison on VBench2.0 using all 18 metrics.

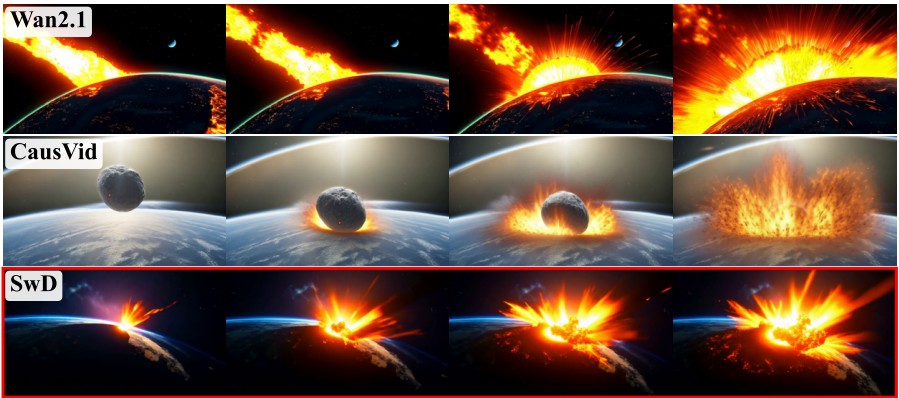

*A breathtaking image of a meteor colliding with the surface of a planet, with bright...*

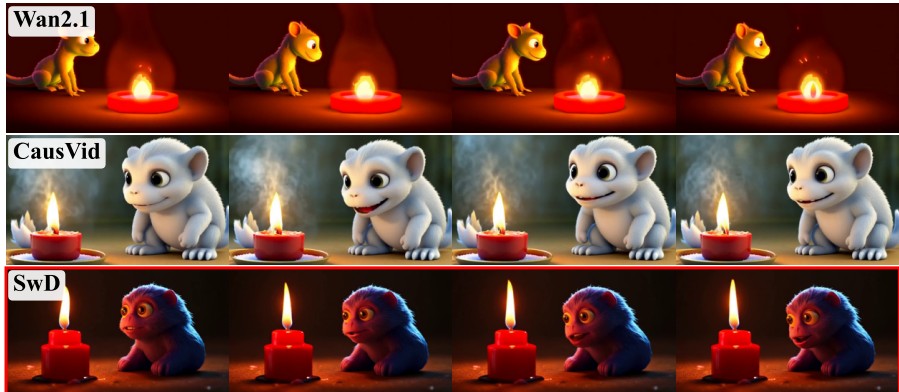

*Animated scene features a close-up of a short fluffy monster kneeling beside a...*

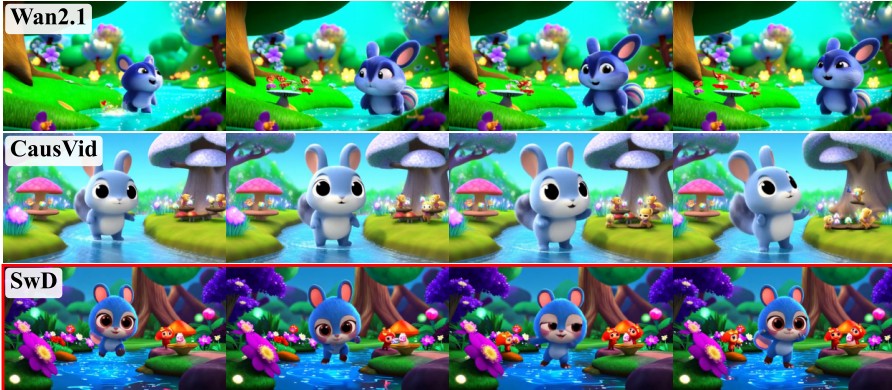

*3D animation of a small, round, fluffy creature with big, expressive eyes explores a...*

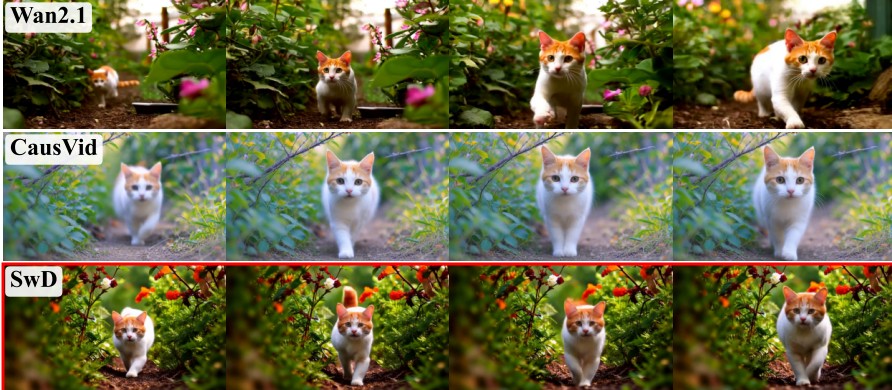

*A white and orange tabby cat is seen happily darting through a dense garden, as if...*

Figure 14: Qualitative results of Wan2.1-SwD.

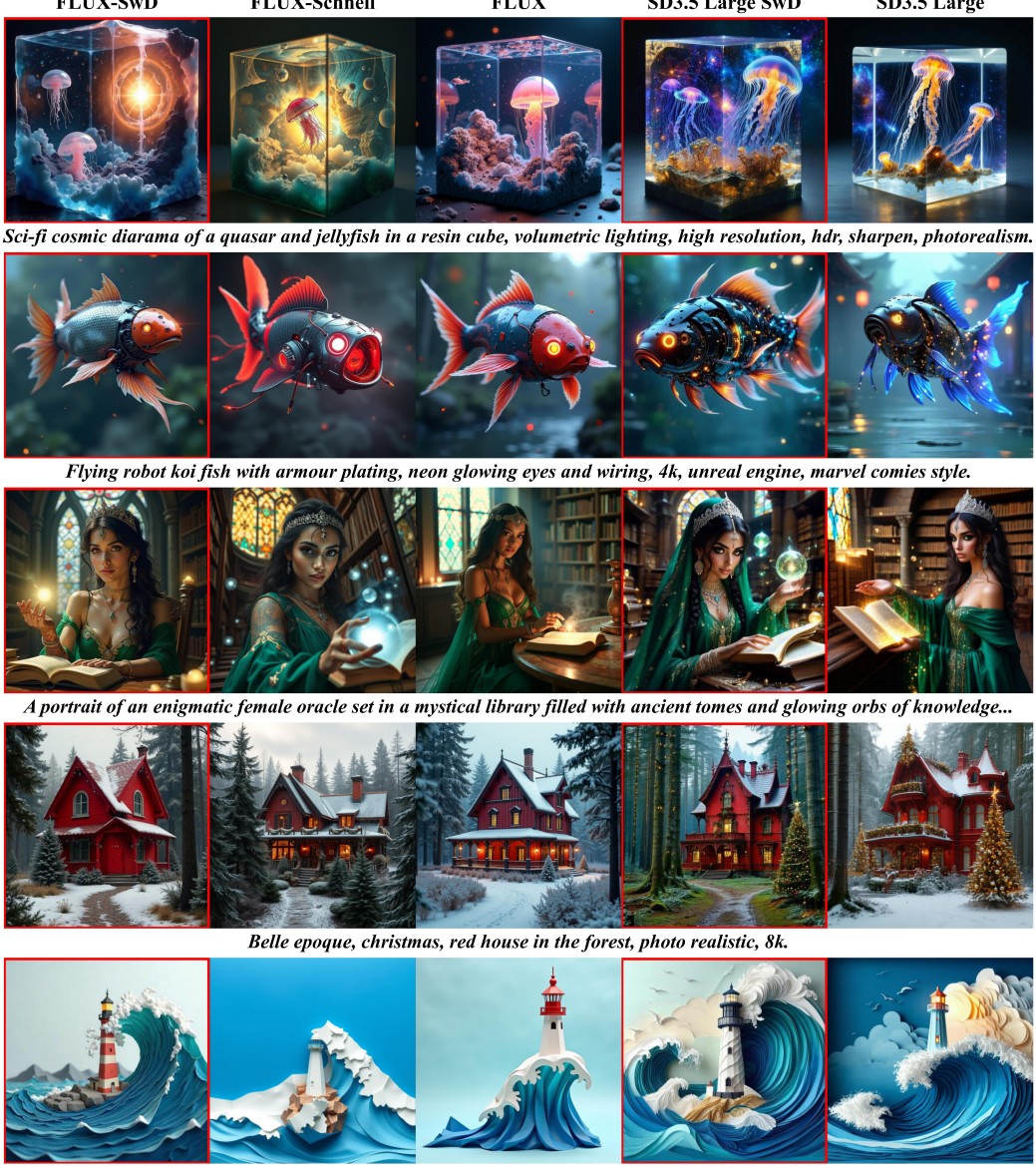

Figure 15: Qualitative results of FLUX-SwD and SD3.5 Large SwD.

| FLUX $\mathcal{L}_{\mathsf{SwD}}$ | FLUX $\mathcal{L}_{\mathsf{MMD}}$ | SD3.5 Medium $\mathcal{L}_{\mathsf{SwD}}$ | SD3.5 Medium $\mathcal{L}_{\mathsf{MMD}}$ |
|---|---|---|---|

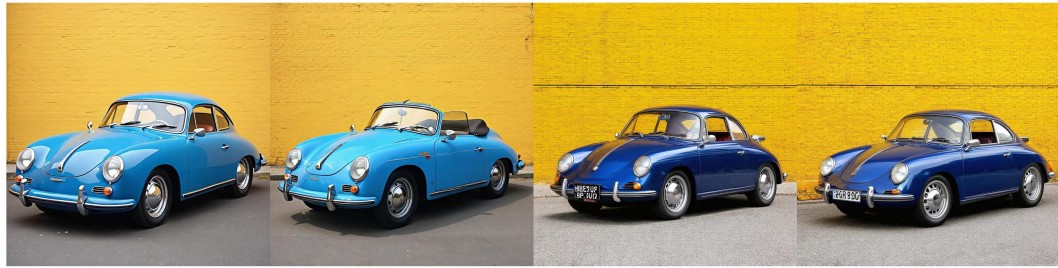

*A close-up high-contrast photo of Sydney Opera House sitting next to Eiffel tower...*

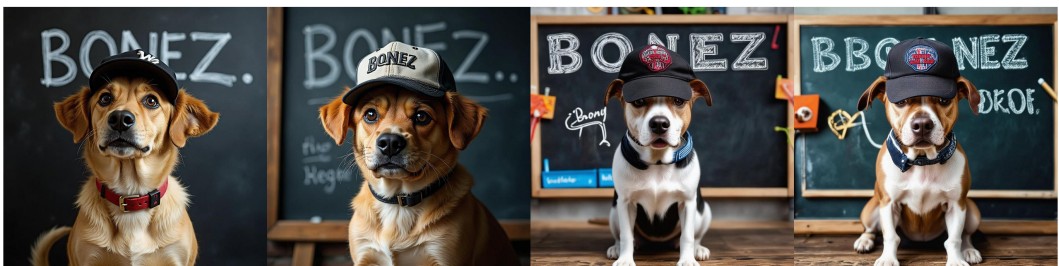

*A blue Porsche 356 parked in front of a yellow brick wall.*

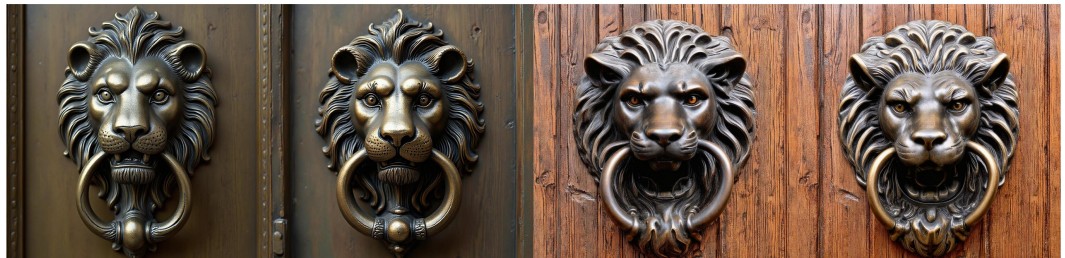

*A dog wearing a baseball cap backwards and writing BONEZ on a chalkboard.*

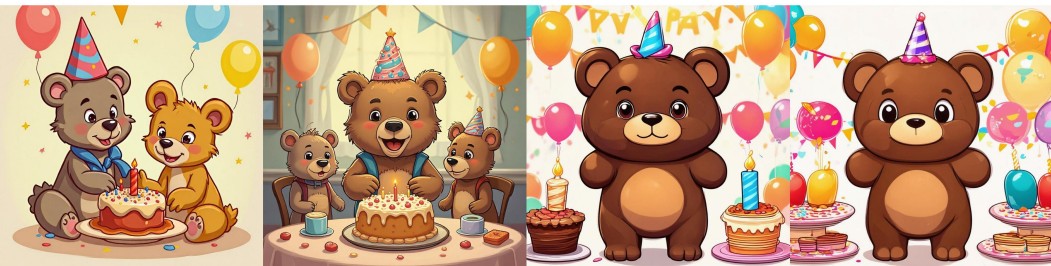

*A doorknocker shaped like a lion's head.*

*A cartoon of a bear birthday party.*

Figure 16: Qualitative comparisons of SwD trained with the full $\mathcal{L}_{\mathsf{SwD}}$ loss against the ones trained with $\mathcal{L}_{\mathsf{MMD}}$ alone. $\mathcal{L}_{\mathsf{MMD}}$ in isolation produces competitive few-step models.

| Scale-wise, 6 steps | Scale-wise, 4 steps | Full-scale, 2 steps | Full-scale, 4 steps | Full-scale, 6 steps |

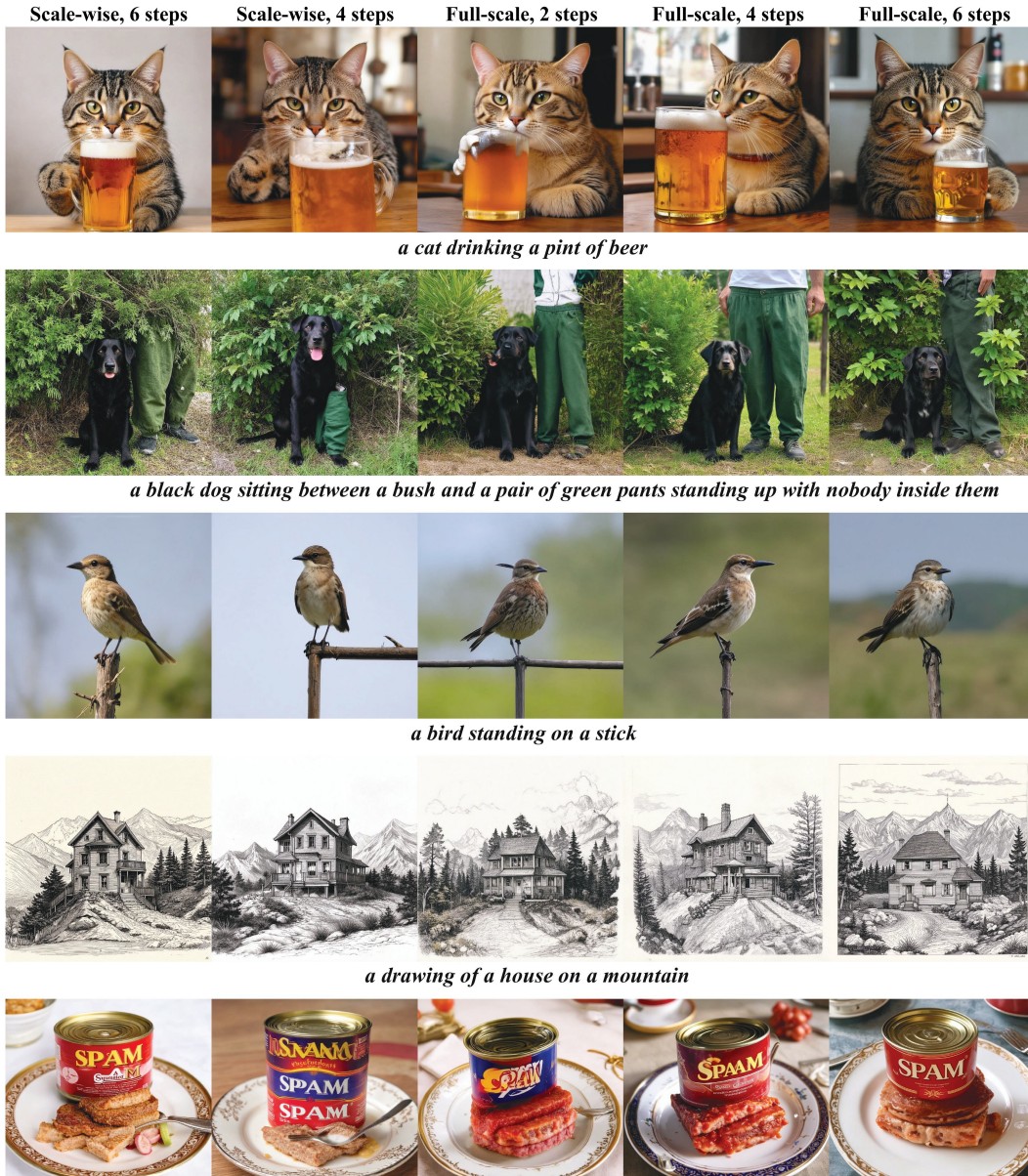

*a cat drinking a pint of beer*

*a black dog sitting between a bush and a pair of green pants standing up with nobody inside them*

*a bird standing on a stick*

*a drawing of a house on a mountain*

*a can of Spam on an elegant plate*

Figure 17: Qualitative examples of image generations using scale-wise and full-resolution SD3.5 Medium SwD variants for different generation steps.

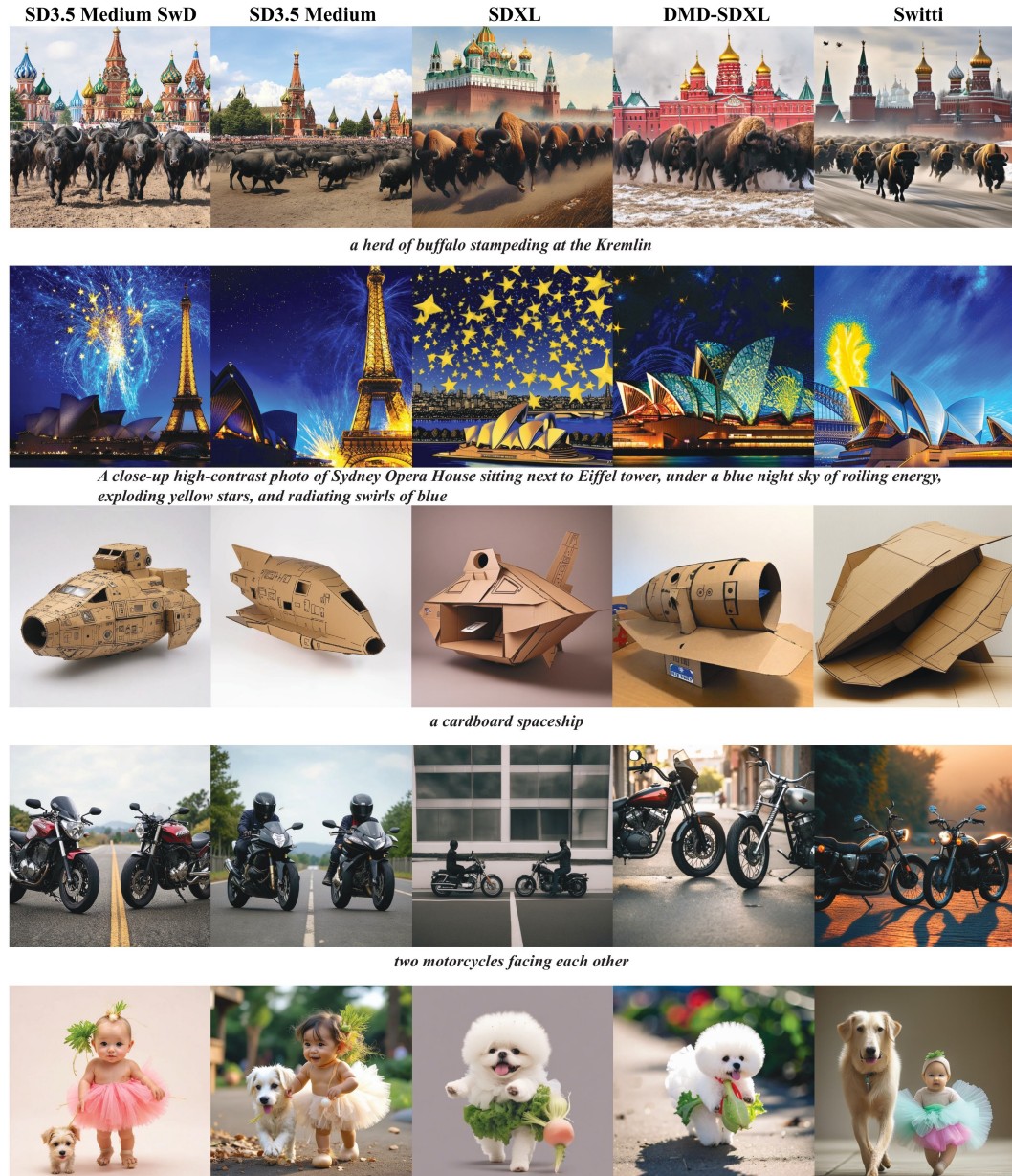

Figure 18: Qualitative comparison of SD3.5 Medium SwD against the models of the similar size.

| SDXL-SwD | DMD2 | SDXL | SDXL-Turbo | Hyper-SD |
|---|---|---|---|---|

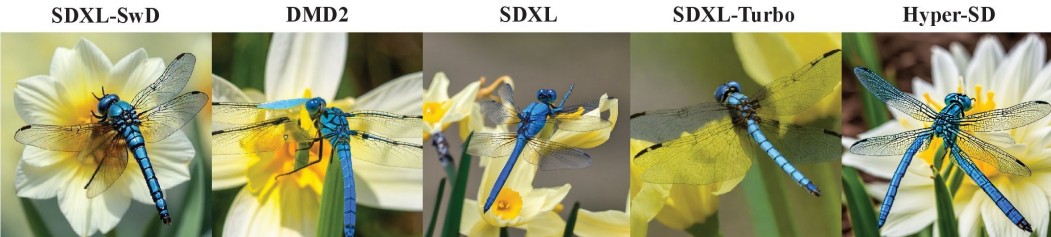

*a close-up of a blue dragonfly on a daffodil*

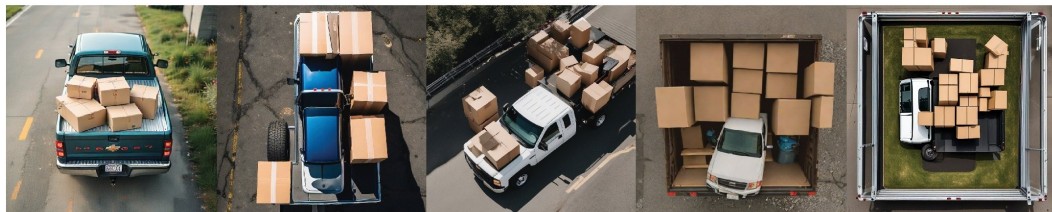

*an overhead view of a pickup truck with boxes in its flatbed*

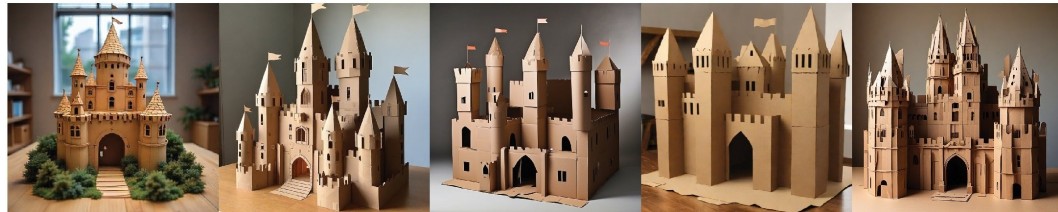

*A castle made of cardboard.*

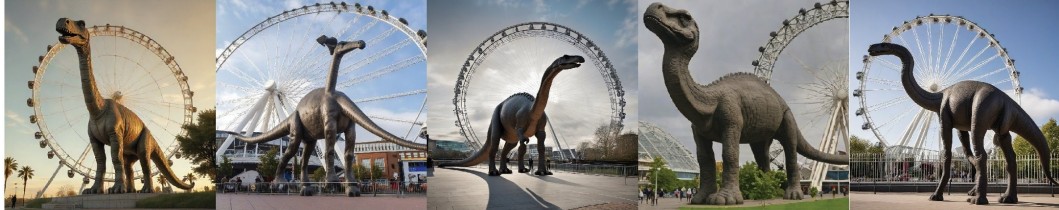

*a diplodocus standing in front of the Millennium Wheel*

Figure 19: Qualitative comparison of SDXL-SwD against the SDXL-based alternatives.

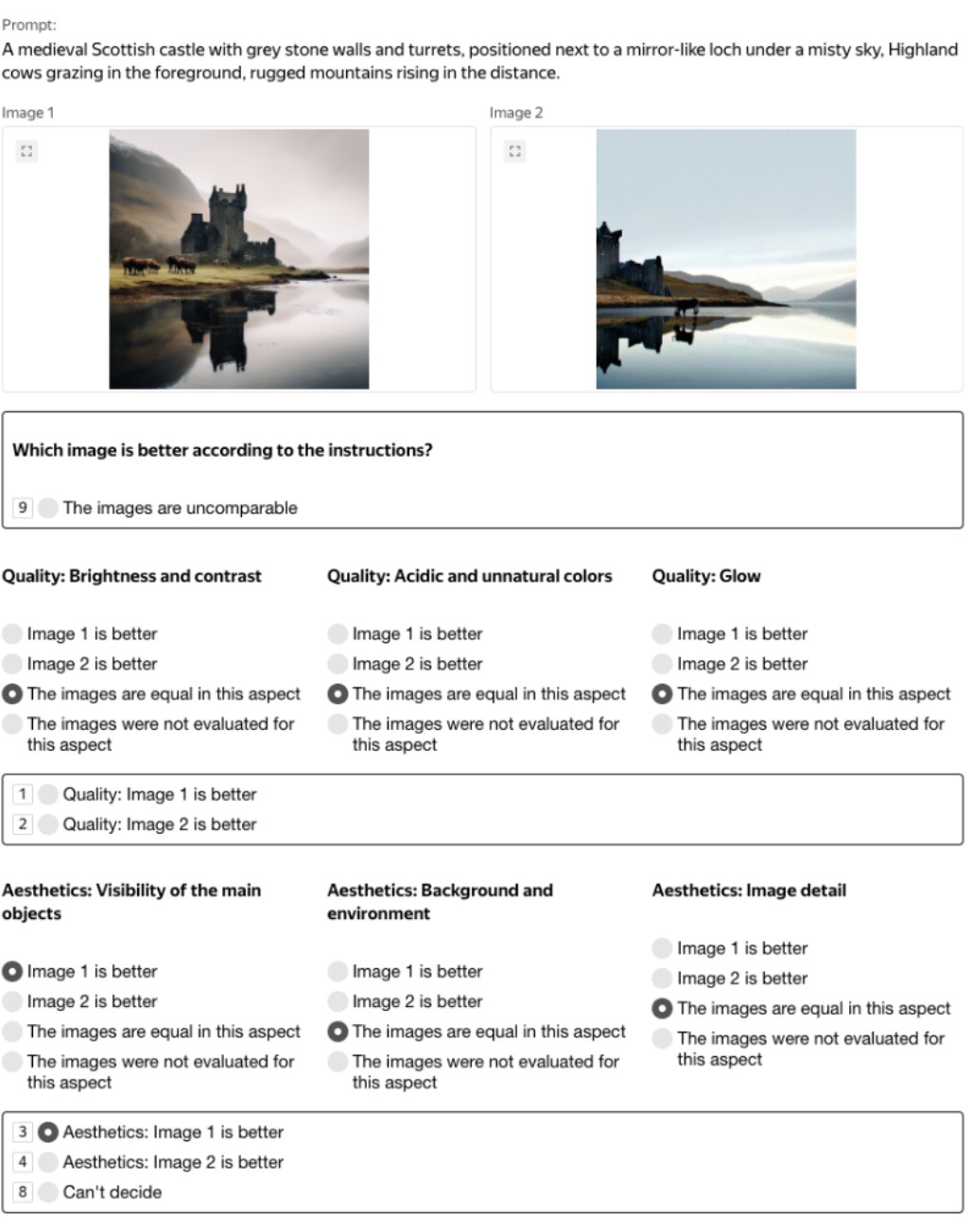

Figure 20: Human evaluation interface for aesthetics.

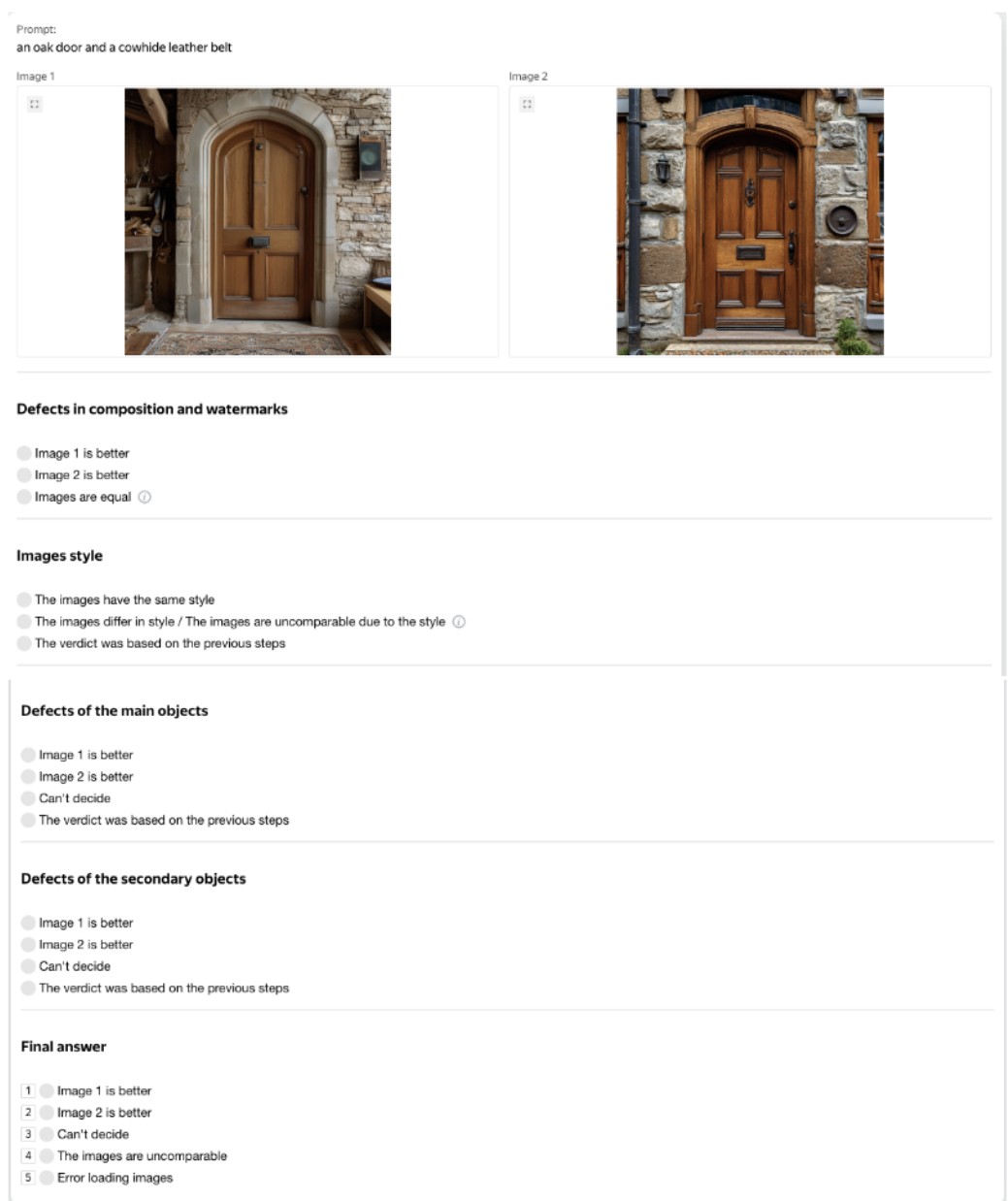

Figure 21: Human evaluation interface for defects.

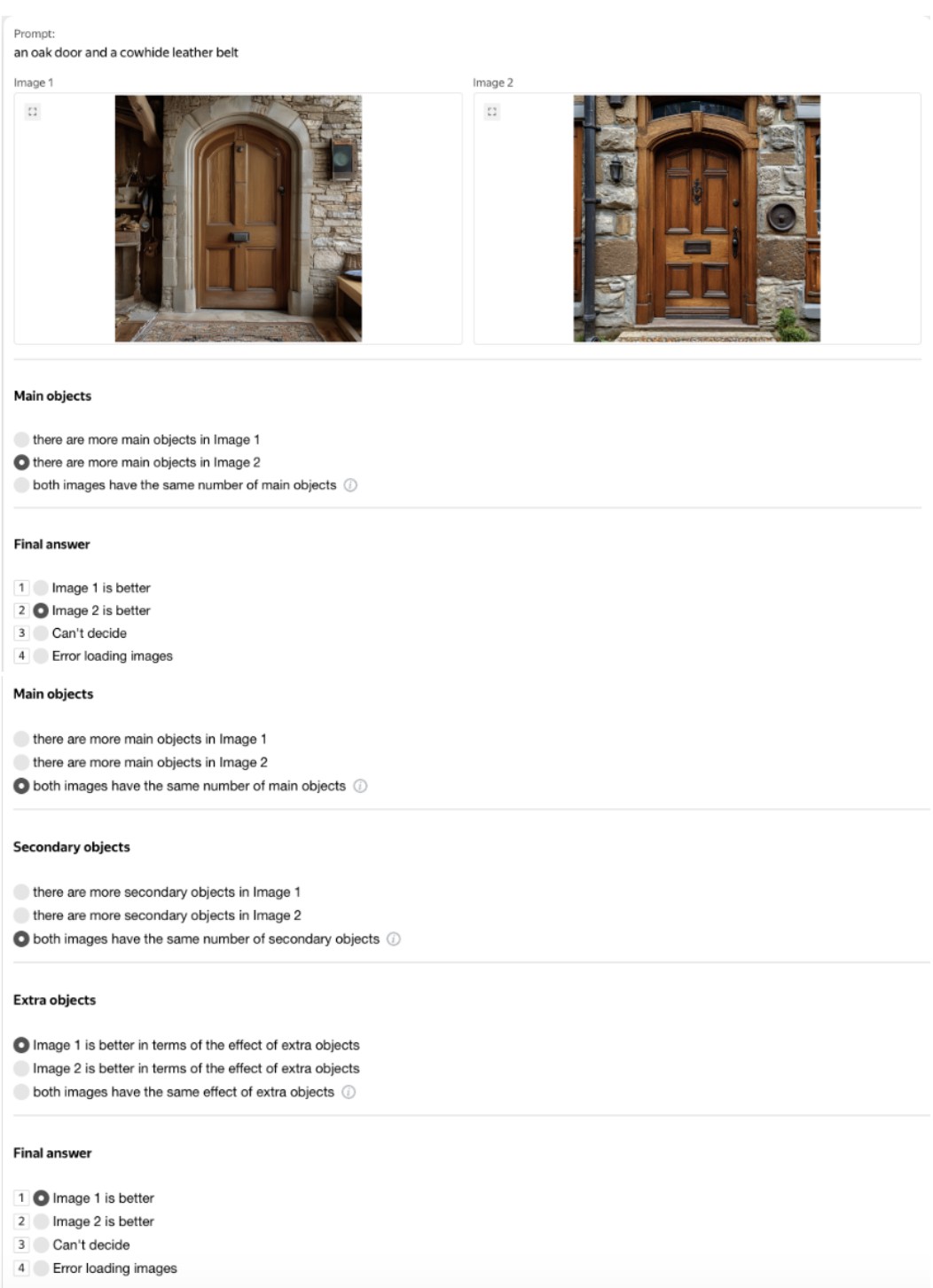

Figure 22: Human evaluation interface for relevance.

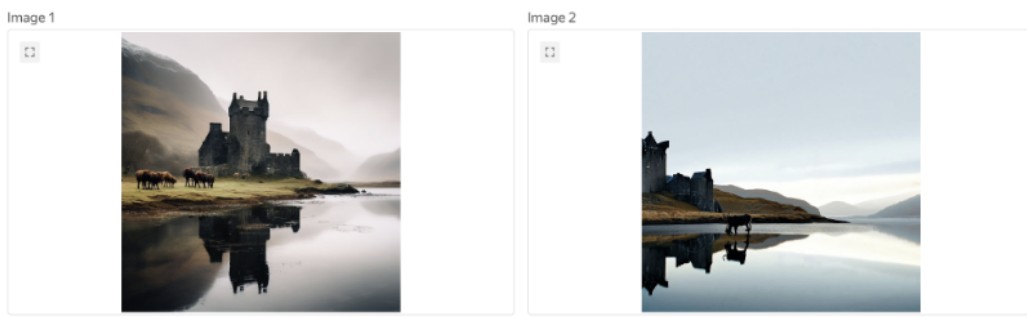

Figure 23: Human evaluation interface for complexity.

