# OpenReview forum: "Scale-wise Distillation of Diffusion Models"
_ICLR.cc/2026/Conference — ICLR 2026 Poster_

### Official Review · Reviewer_8Ahm · 2025-10-30

**Soundness:** 2
**Presentation:** 2
**Contribution:** 2
**Rating:** 4
**Confidence:** 4

**Summary:**

This paper proposes Scale-wise Distillation (SwD), a method that distills a diffusion model into a single model capable of progressively increasing resolution while generating images with only a few sampling steps.

The key ideas are:
(i) Spectral analysis, which shows that in the early high-noise stages, high-frequency components are largely suppressed, thus computations can be performed at lower spatial/temporal resolutions to save cost;
(ii) A progressive sampling and training procedure, where at each step the resolution is increased, and the previous step’s
is upsampled → re-noised to serve as the input for the next step;
(iii) A simple distribution matching loss that aligns teacher and student distributions in the DM feature space using Maximum Mean Discrepancy (MMD), particularly a linear-kernel variant (LMMD).

**Strengths:**

-  **Method motivation**: Through RAPSD-based analysis, the paper convincingly demonstrates why low-resolution processing is safe during high-noise stages (Fig. 1). This motivation is appropriate as a methodological rationale; however, similar motivations have actually been used several times before from the diffusion efficiency literature. See Weakness.

- **Solid experiments**: The approach is also extended to both text and video domains. The comparison of efficiency in Tables 4 and 5 is very appropriate, and the performance comparison is clearly shown in Figure 7. Under the same number of steps, SWD and the full model seem to have comparable performance, but SWD shows better efficiency.

**Weaknesses:**

- Recently, many diffusion efficiency studies have discussed that it is reasonable to focus on low-frequency components at time steps close to the noise. For example, [1] uses a transformer with a larger patch size at earlier (noise-near) time steps, and [2] proposes a method to better capture low-frequency information at each time step. It would be better if the motivation of this work were discussed in connection with these findings.

- My greatest concern is whether the proposed method can effectively capture high-frequency details. Since the method focuses largely on low-frequency content except at the final step, this potentially poses a limitation. Moreover, because large-resolution images are involved, it may be difficult for the metrics to properly evaluate whether the high-frequency information has been preserved. I strongly recommend incorporating the measurement of FID-Patch as used in the SDXL‑Lightning paper to comprehensively assess fine detail preservation.


[1] (CVPR 25) FlexiDiT: Your Diffusion Transformer Can Easily Generate High-Quality Samples with Less Compute

[2] (CVPR 25) Autoregressive Distillation of Diffusion Transformers

[3] SDXL-Lightning: Progressive Adversarial Diffusion Distillation

**Questions:**

See weakness.

---

> ### Author Response · Authors · 2025-11-22
> **Reply to Official Review by Reviewer 8Ahm**
>
> We sincerely appreciate the reviewer for insightful feedback and thoughtful questions.
>
> **W1: Discussion on FlexiDiT and Autoregressive Distillation.**
>
> We thank the reviewer for suggesting these works, and we included them in our related work section.
>
> FlexiDiT falls within the line of progressive DMs discussed in (L090–L100) and is also inspired by the spectral autoregression perspective of image-space DMs, explored in earlier works [1,2]. In contrast, our work is the first to investigate this perspective in *VAE latent spaces* and to further extend it to the temporal dimension in the video domain. Then, our work introduces a scale-wise diffusion *distillation* framework that, to our knowledge, is the first to enable progressive generation in few-step models. In future work, it would be promising to employ the FlexiDiT idea of using larger patch sizes at different scales rather than explicitly using upscaling.
>
> ARD improves distillation by introducing a causal mask over previous timesteps. We believe this method is largely orthogonal to ours and potentially complementary. It would be interesting to combine it with our approach in future work.
>
> [1] Rissanen et al. Generative modelling with inverse heat dissipation, ICLR2023
>
> [2] Dieleman. Diffusion is spectral autoregression, 2024
>
> **W2: Patch FID evaluation.**
>
> We agree with the reviewer that the question regarding high frequency preservation is important. To address this, we compare SwD with its full-resolution counterpart and provide Patch FID results on MJHQ30K below as requested. Note that our models generate 1024x1024 images while MJHQ30K also contains 1024x1024 images, so the evaluation is valid. We observe that SwD provides similar Patch FID values compared to the full-resolution counterparts.
>
> | Model       | Steps |  SwD |  Full-resolution|
> | ----------- | ------| ----- | --------------------------- |
> | SD3.5-M     | 6 |     20.1    |            21.1             |
> | SD3.5-L     | 4 |     20.4    |            21.9             |
> | FLUX        | 4 |     18.7    |            18.9             |
>
>
> We also evaluate a state-of-the-art no-reference IQA metric (CLIP-IQA $\uparrow$) on image patches to further confirm high-frequency detail preservation. As shown in the table below, SwD again achieves similar scores to the full-resolution models, indicating that fine-grained details are not noticeably degraded.
>
> | Model       | Steps | SwD | Full-resolution|
> | ----------- | ------|----- | --------------------------- |
> | SD3.5-M     | 6 |    0.85    |            0.85             |
> | SD3.5-L     | 4 |    0.88    |            0.88             |
> | FLUX        | 4 |    0.80    |            0.79             |
>
> We included all these results in Appendix D.

---

### Official Review · Reviewer_h2GD · 2025-10-31

**Soundness:** 3
**Presentation:** 3
**Contribution:** 3
**Rating:** 6
**Confidence:** 5

**Summary:**

This paper advances diffusion distillation by introducing a scale-wise few-step diffusion model. Instead of operating entirely at a fixed resolution, the model begins at a lower scale and progressively increases to the final resolution, similar to standard few-step diffusion approaches. Additionally, the authors propose a Maximum Mean Discrepancy (MMD)-based distillation loss to complement the existing DMD and GAN losses, further improving training effectiveness. Experiments conducted on various architectures—including SD 3.5 and FLUX for image generation (evaluated on COCO2014 and MJHQ) and WAN 2.1 for video generation (evaluated on VBench 2.0)—demonstrate the effectiveness of the proposed method.

**Strengths:**

1. The paper presents a detailed spectral analysis of the latent spaces across different diffusion models, offering valuable insights that motivate the first contribution—scale-wise distillation.

2. The proposed distillation framework is evaluated on multiple diffusion models, covering both image and video generation tasks. The results demonstrate the potential of scale-wise distillation to achieve higher efficiency while maintaining performance comparable to existing distillation methods.

3. The paper is clearly written and well-organized, with logical section flow and well-explained experimental setups that effectively support its main conclusions.

**Weaknesses:**

1. The experiment on different upsampling strategies (lines 192–212) lacks evaluations with alternative scale configurations (e.g., from/to 32, 80, or 96), which would provide a more comprehensive and reliable analysis.

2. The paper does not explain why the temporal dimension of SwD does not contribute to performance improvement when applied to Wan 2.1 (lines 375–377).

3. The human preference study charts in Figures 6, 7, and 8 lack sufficient descriptive captions or visual annotations, reducing interpretability.

4. In Table 3, the performance of SwD on 8B- and 12B-scale models shows only marginal gains over other distilled variants—for instance, SD3.5-L-Turbo (0.71 vs. 0.70) and FLUX-Schnell (0.71 vs. 0.69). Moreover, the human preference study (Figure 6) indicates that SD3.5-L-SwD and FLUX-SwD are outperformed by Turbo-L and FLUX-Schnell in most aspects, calling into question the practical advantage of SwD at similar scales.

5. The paper omits comparisons with other few-step video diffusion models, such as Video LCM [1], T2V-Turbo [2], and MagicDistillation [3], which are relevant baselines.

6. There is no discussion of Jenga [4], which shares a similar conceptual idea of adjusting scale with timestep (smaller scales for larger timesteps and larger scales for smaller timesteps).

7. Table 3 lacks a description of the scale schedule corresponding to each timestep, making the setup difficult to reproduce.

[1] https://arxiv.org/abs/2312.09109

[2] https://arxiv.org/abs/2410.05677

[3] https://arxiv.org/abs/2503.13319

[4] https://arxiv.org/abs/2505.16864

**Questions:**

1. Why does the GenEval result for Infinity reported in this paper (0.69 in Table 3) differ from the value reported in the original Infinity paper (0.73)?

2. In Equation (2), how are the two presented kernels applied in practice?

3. Are the models initialized from pretrained weights or trained from scratch?

4. Could the authors provide additional explanation for why the temporal dimension in SwD does not lead to performance improvements in Wan 2.1? (lines 375–377)

---

> ### Author Response · Authors · 2025-11-22
> **Reply to Official Review by Reviewer h2GD (Part 1)**
>
> We greatly appreciate the reviewer’s constructive feedback and address the concerns below.
>
> **W1 | Additional results for different upscale ratios (Table 1).**
>
> Below, we extend Table 1 with $32\times32 \rightarrow 128\times128$ and $96\times96 \rightarrow 128\times128$ setups. Upscaling the noised latents provides poor performance across all upscale factors. For upscaling $x_0$ predictions, as expected, lower upscale ratios consistently lead to better results, since the latents are less distorted by upscaling. We added the extended table to Appendix E.
>
> |                           Configuration                              | FID t=400 | FID t=600 | FID t=800 |
> | ---------------------------------------------------------------------| --------- | --------- | --------- |
> | $x_0^{128\times128} \rightarrow x_t^{128\times128}$                        |    9.2    |    10.3   |    13.7   |
> | $x_0^{32\times32} \rightarrow x_0^{128\times128} \rightarrow x_t^{128\times128}$ |    90.2   |    40.8   |    16.6   |
> | $x_0^{32\times32} \rightarrow x_t^{32\times32} \rightarrow x_t^{128\times128}$   |    217    |    298    |    361    |
> | $x_0^{64\times64} \rightarrow x_0^{128\times128} \rightarrow x_t^{128\times128}$ |    30.9   |    18.8   |    14.7   |
> | $x_0^{64\times64} \rightarrow x_t^{64\times64} \rightarrow x_t^{128\times128}$   |    122    |    223    |    327    |
> | $x_0^{96\times96} \rightarrow x_0^{128\times128} \rightarrow x_t^{128\times128}$ |    18.8   |    14.3   |    14.3   |
> | $x_0^{96\times96} \rightarrow x_t^{96\times96} \rightarrow x_t^{128\times128}$   |    32.8   |    58.9   |    120    |
>
>
>
> **W2/Q4 | Clarifying the conclusion on Wan-SwD gains in L375–377.**
>
> We apologize for the confusing formulaton. We meant there that SwD across both dimensions does not degrade compared to the spatial-only variant in quality. This supports our claim that scale-wise distillation can be effectively performed along both dimensions. Note that it still provides acceleration, although it is less pronounced since reducing along the tempotal dimension leads to linear speedup, while along the spatial one - quadratic. Also, video VAEs runtimes become noticeable in extremely few-step settings, also hindering the overall gains.
>
> **W3: human study lacks sufficient descriptive captions or visual annotations.**
>
> In Appendix H, we describe our human study pipeline and provide the exact user interfaces used for each evaluated aspect in Figures 19, 20, 21. These interfaces include the full set of questions presented to annotators during evaluation. Please let us know if there is any additional information that would be helpful to include.
>
> **W4/1: Marginal GenEval gains on 8B-12B models.**
>
> We would like to note that GenEval measures a model’s ability to follow the prompt accurately and does not assess image aesthetics, complexity, or fidelity. Therefore, for distilled models, the desirable outcome is to preserve the teacher model’s level of textual relevance, which we consistently observe for all SwD variants, as supported by both GenEval results and our human study. Meanwhile, the metrics designed to evaluate overall image fidelity (PickScore, HPSv3, and ImageReward) show that our models achieve consistent and noticeable improvements.
>
> **W4/2: Human study indicates that SD3.5-L-SwD and FLUX-SwD are outperformed by Turbo-L and FLUX-Schnell in most aspects**
>
> We respectfully disagree with this claim. On the contrary, our distilled models outperform FLUX-Schnell and Turbo-L by a large margin in both aesthetics and complexity, and perform on par with them (within standard deviation) in terms of defects and textual relevance. Moreover, SwD still offers $\sim2{\times}$ faster inference over these baselines.
>
> **W5: No comparisons with few-step video DMs (Video LCM, T2V-Turbo, and MagicDistillation)**.
>
> Although requested, we were unable to find any publicly released models for VideoLCM or MagicDistillation. To address the reviewer’s concern, we compare against T2V-Turbo, for which released models are available.
>
> We also note that T2V-Turbo is built on VideoCrafter2, whereas our method is based on Wan2.1-1.3B. In addition, the released T2V-Turbo model supports only 1-second video generation, while Wan2.1 supports 5 seconds, making the models largely uncomparable. Nevertheless, we provide the results in the table below and observe that our models substantially outperform T2V-Turbo-V2.
>
> | Model | VisionReward↑ | VideoReward↑ | VBench2↑ |
> |---|---|---|---|
> T2V-Turbo-V2 |   0.021     |    0.90    |    43.89 |
> Wan-SwD      |   **0.064**     |    **6.27**    |    **53.22** |

---

> ### Author Response · Authors · 2025-11-22
> **Reply to Official Review by Reviewer h2GD (Part 2)**
>
> **W6 | Missing discussion of Jenga.**
>
> We thank the reviewer for suggesting this work and we cited it in our related work section. Jenga is a progressive video DM that shares similar ideas to the progressive image DMs discussed in our related work (L090-L100). In contrast, our work proposes a novel scale-wise *distillation* method which goes beyond these prior approaches.
>
> **W7 | Table 3 lacks a description of the scale schedules.**
>
> We would like to note that all our timestep and scale schedules for each model presented in Table 3 are provided in Appendix D (L893–904) and referenced in the main text (L318).
>
> **Q1 | Infinity GenEval.**
>
> The reported 0.73 is provided with their prompt rewriter, while original GenEval is 0.69. Please see the comment in the Infinity repository (https://github.com/FoundationVision/Infinity?tab=readme-ov-file#infinity)
>
> **Q2 | In Equation (2), how are the two presented kernels applied in practice?**
>
> We extract the intermediate transformer token features for both the source and target images. Equation (2) corresponds to MMD with a linear kernel: for each image, we average the feature vectors across all tokens and then compute the MSE between these average tokens.
>
> For the RBF-kernel version, we compute all pairwise L2 distances between the source and target token features to obtain the terms $k(x, x') = e^\frac{\lVert x-x'\rVert^2}{2\sigma^2}$, $k(y, y') = e^\frac{\lVert y-y'\rVert^2}{ 2\sigma^2}$ and $k(x, y) = e^\frac{\lVert x-y\rVert^2}{2\sigma^2}$, where $x$ and $y$ correspond to source and target image tokens. Then, MMD is calculated using Equation (1).
>
> **Q3 | Student Initialization.**
>
> All models are initialized with corresponding pretrained teacher models.

---

### Official Review · Reviewer_cK3V · 2025-10-31

**Soundness:** 3
**Presentation:** 3
**Contribution:** 3
**Rating:** 6
**Confidence:** 4

**Summary:**

This paper addresses the inefficiency of few-step diffusion distillation methods, which redundantly compute all steps at full resolution. The authors propose two main contributions.First, Scale-wise Distillation (SwD), a framework where a single generator progressively increases its operating resolution during the few-step sampling process. This avoids unnecessary computation in early, noisy steps, where high-frequency detail is absent. Second, a simple and effective MMD-based distillation loss ($\mathcal{L}_{MMD}$) that matches student and teacher feature distributions. This loss is shown to be a highly competitive baseline on its own and significantly speeds up training. Applied to SOTA models (SD3.5, FLUX, Wan2.1), SwD provides ~2-3x speedup over full-resolution models at the same step count and achieves significantly higher quality under the same computational budget

**Strengths:**

1. The core idea of unifying progressive generation with few-step distillation (SwD) is novel and elegant. The motivation is strong, grounded in a solid spectral analysis (Section 3) of VAE latents for both images and video;
2. The paper introduces a simple MMD-based loss that is surprisingly powerful. Ablations (Table 6) show it performs competitively on its own while being remarkably efficient. As it requires no extra trainable models (unlike GAN or DMD losses), it enables >7x faster training iterations (Table 5), making it a highly valuable standalone contribution;
3. The experiments are comprehensive. The key comparison in Section 5.2 (Figure 7, Tables 7-8) clearly demonstrates SwD's superiority: at an equivalent compute cost (e.g., 4-step SwD vs. 2-step Full-res), SwD produces significantly better-quality images with fewer defects . The main results (Table 3, Figure 6) show SOTA performance, even outperforming teacher models in human preference;

**Weaknesses:**

1. The framework's performance depends on the co-design of the timestep schedule $[t_i]$ and the scale schedule $[s_i]$. While the authors provide the schedules used (Appendix D), the paper offers limited intuition on the methodology for finding these optimal schedules or the model's sensitivity to them.
2. While the $ \mathcal{L} _ {MMD} $ only variant is exceptionally simple, the full $ \mathcal{L} _ {SwD} $ objective used to achieve the absolute SOTA results ($\mathcal{L} _ {MMD}+\mathcal{L} _ {DMD}+\mathcal{L} _ {GAN}$) inherits the training complexity of methods like DMD2 (e.g., training a "fake" DM). However, the paper wisely presents the $\mathcal{L} _ {MMD}$-only path as a highly effective, simpler alternative.

**Questions:**

see weaknesses

---

> ### Author Response · Authors · 2025-11-22
> **Reply to Official Review by Reviewer cK3V**
>
> We deeply appreciate the reviewer for the positive assessment and thoughtful feedback. Below, we answer the raised questions.
>
> **W2 | Clarifications on finding optimal scale and timestep schedules**
>
> In our work, we did not perform extensive tuning of either the timestep or scale schedules, and the same schedules are shared across different models. The timestep schedules were obtained by starting from the standard few-step schedules induced by the teacher scheduler and slightly shifting them toward noisier timesteps to reduce upscaling artifacts. The scale schedules were set to increase uniformly from 32×32 or 64×64 to 128×128, motivated by achieving noticeable acceleration gains.
>
> We also experimented with reasonable variations of timestep and scale schedules for both the 4-step and 6-step setups and did not observe any significant differences. This suggests that the method is not highly sensitive to the specific schedule choice. We have expanded our discussion of schedule selection in L258–262.
>
> **W2 | Full objective used to achieve the absolute SOTA results  inherits the training complexity of methods like DMD2. However, the paper wisely presents the MMD-only path as a highly effective, simpler alternative.**
>
> We agree with this point and explicitly highlight it in the paper. Threfore, we encourage future work to further investigate the MMD objective as a potential way to fully eliminate the need for the expensive DMD2 pipeline. We believe it would be promising to explore combining the MMD loss with simple knowledge distillation or consistency-based approaches.

---

### Official Review · Reviewer_iUZm · 2025-10-31

**Soundness:** 2
**Presentation:** 2
**Contribution:** 2
**Rating:** 4
**Confidence:** 5

**Summary:**

The paper introduces Scale-wise Distillation (SwD), which progressively increases latent resolution during few-step diffusion distillation.
Motivated by spectral analysis showing that high-noise latents mainly contain low-frequency content, the method reduces redundant computation at early steps.
An additional MMD-based distillation loss is proposed and combined with DMD, showing competitive results on SD3.5, FLUX, and Wan2.1 models.

**Strengths:**

* Clear and well-motivated idea linking noise level and latent frequency spectrum.
* Practical framework that integrates smoothly with existing distillation methods.
* The proposed MMD loss is easy to use, and effective even alone.
* Good results with notable speedups for both text-to-image and text-to-video generation.
* The writing is clear and the figures (e.g., Figure 1 spectral plots) effectively communicate the intuition behind the framework.

**Weaknesses:**

1. **Limited and possibly unfair baseline comparisons (SDXL case):**
Table 3 compares SwD results on SD3.5 and FLUX but omits SwD results on SDXL, even though Appendix B (Figure 9) shows that SDXL experiments were performed. For a fair evaluation, the authors should include SDXL-SwD and compare it directly to DMD2-SDXL and SDXL-Turbo. Moreover, other recent open-source distillation baselines such as Hyper-SD [Hyper-SD: Trajectory Segmented Consistency Model for Efficient Image Synthesis] should be incorporated.
Given that the paper emphasizes the synergy between MMD + DMD losses, fair quantitative evidence across identical model backbones is essential to support the claimed superiority.

2. **Insufficient validation for FLUX and Wan distillations:**
The comparison for FLUX and Wan 2.1 models is not convincing, as no competing baselines are provided. Stronger open-source baselines such as Hyper-FLUX, CausVid, or LightX2V (all publicly available and Wan-based distillation frameworks) should be included. Without these comparisons, it is difficult to judge the SwD’s efficiency and quality againest other distillation methods.

3. **a minor issue** :
In Table 3, the number of function evaluations (NFE) or diffusion steps used for each model is unclear, making speed–quality trade-offs hard to assess.


4. **Relationship between MMD loss and prior progressive distillation methods (ADD/LADD) is not well clarified:**
The MMD loss seems similar with the progressive distillation in ADD/LADD with different training objective. (constrain intermediate latent representations via feature alignment.) The paper should discuss more explicitly:
* Whether the training procedure of “MMD-only distillation” is otherwise identical to ADD or DMD pipelines aside from the loss definition.
* What specific advantages (e.g., stability, generalization) MMD introduces beyond computational simplicity.

**Questions:**

Please see the part of Weaknesses.

---

> ### Author Response · Authors · 2025-11-22
> **Reply to Official Review by Reviewer iUZm (Part 1)**
>
> We sincerely thank the reviewer for careful reading and valuable feedback. We address the concerns and questions below.
>
>
> **W1 | SDXL-SwD and comparison with Hyper-SD.**
>
> We agree with the reviewer that evaluating SwD on SDXL would provide a better understanding of its applicability across different architectures. To this end, we distill SDXL using a [64,80,96,128] scale schedule and report the quantitative results below. The results show that SDXL-SwD outperforms Hyper-SD, DMD2, and Turbo in most cases, while also being nearly twice faster. We added these results to Table 3 of the main paper and also included the qualitative comparison in Figure 19.
>
> | Model | NFE | Image size | Latency | COCO 30K | | | | MJHQ 30K | | | | |
> |---|---|---|---|---|---|---|---|---|---|---|---|---|
> | | | | | PS↑ | HPSv3↑ | IR↑ | FID↓ | PS↑ | HPSv3↑ | IR↑ | FID↓ | GenEval↑ |
> SDXL-Turbo | 4 | 512 | 0.12 | 22.6 |  10.0 |  0.83 |  17.5  | 21.3  | 9.6  |  0.84 |  15.4  |   0.55|
> SDXL-DMD2 | 4 | 1024 | 0.20 | 22.8  | 12.0  | 0.87  | **14.1**  | **21.6** |  10.1  | 0.86 |  **8.3**   |   **0.58**|
> Hyper-SD | 4 | 1024 | 0.20 | 22.8  | 11.0|  0.90  | 20.1 |  **21.6**  | 10.1  | 0.94 |  14.7  |   0.55|
> SDXL-SwD | 4 | 1024 | **0.11** | **22.9** | **12.4** | **0.95** | 21.3 | **21.6** | **10.3** | **0.97** | 15.1 | 0.57
>
> **W2 | Additional FLUX and Wan2.1 baselines.**
>
> As requested, we first provide the quantitative comparison with the best performing Hyper-FLUX setup below. FLUX-SwD largely outperforms Hyper-FLUX in terms of GenEval, HPSv3, PickScore and ImageReward. Moreover, FLUX-SwD is more than $4{\times}$ faster than Hyper-FLUX. We added these results to Table 3.
>
> | Model | NFE | Latency | COCO 30K | | | | MJHQ 30K | | | | GenEval↑ |
> |---|---|---|---|---|---|---|---|---|---|---|---|
> | |  |  | PS↑ | HPSv3↑ | IR↑ | FID↓ | PS↑ | HPSv3↑ | IR↑ | FID↓ |
> Hyper-FLUX   |  8  | 2.75 | 23.0 |  12.4 |  0.94  | **24.2**  | 21.7 |  10.9  | 0.85  | 14.9   |  0.61|
> FLUX-SwD     |  4  | **0.72** | **23.1**  |  **14.6**   | **1.14**  | 26.4  |  **21.9**  |  **11.6**   | **1.06**  |  **14.4**   |   **0.71** |
>
> Regarding the comparisons of Wan2.1-SwD with CausVid and LightX2V, we would first like to highlight the following points:
>
> - LightX2V provides only distilled Wan2.1-14B models, whereas all of our experiments were conducted with Wan2.1-1.3B. Therefore, we believe this comparison would be unfair, since the 14B model is inherently much stronger. Unfortunately, our current resources do not allow us to distill 14B-scale video models.
> - Although CausVid is built on Wan2.1-1.3B, it was distilled using the internal high-quality dataset as noted by the authors [1], while all our models are trained on the teacher samples. The CausVid authors emphasize that data quality plays a crucial role in overall model performance. [1] https://github.com/tianweiy/CausVid/issues/28#issuecomment-2826578515
>
> To ensure a fair comparison, we therefore reproduce CausVid for Wan2.1-1.3B using their official implementation and configs, but train it on our dataset. The results are provided below. As can be seen, SwD slightly outperforms CausVid, while achieving ${\sim}2.3\times$ faster inference. We included these results in Table 2.
>
> | Model | Latency | VisionReward↑ | VideoReward↑ | VBench2↑ |
> |---|---|---|---|---|
> CausVid    |  4.2      |   0.042      | 6.21      |  52.31     |
> Wan-SwD    |  **1.8**  |   **0.064**  | **6.27**  |   **53.22** |
>
> To further assess the effectiveness of our method, we finetune the officially released CausVid model to the scale-wise variant by using only the MMD objective and training data sampled from the CausVid model itself. We provide the results below and observe that CausVid-SwD is comparable to the original model while offering ${\sim}2.3\times$ inference speedup. These results also highlight an additional advantage of our approach by allowing adapting already distilled models to the scale-wise variants.
>
> | Model | Latency | VisionReward↑ | VideoReward↑ | VBench2↑ |
> |---|---|---|---|---|
> CausVid      | 4.2  |   0.065     |   **7.12** |   52.98  |
> CausVid-SwD  | **1.8**  |   **0.070** |   **7.13** |   **53.88**   |
>
>
> **W3 | Adding NFEs to Table 3.**
>
> Thanks for the suggestion. We added NFEs for each model to the table.

---

> ### Author Response · Authors · 2025-11-22
> **Reply to Official Review by Reviewer iUZm (Part 2)**
>
> **W4 | MMD-only variant seems similar to prior progressive distillation in ADD/LADD with a different training objective. Is the training procedure of “MMD-only distillation” is identical to ADD or DMD aside from the loss definition?**
>
> We would like to clarify that the methods like ADD/LADD/DMD are not *progressive* in the sense of enabling scale-wise generation. To our knowledge, prior distillaton methods have not explored this degree of freedom, motivating our approach in the first place. Therefore, “MMD-only distillation” is not identical to ADD/DMD aside from the loss definition.  Our main contributions are both the proposed scale-wise distillation framework and the MMD-based distillation loss, which substantially differs from the objectives used in ADD and DMD. Please let us know if there is something that needs to be further clarified.
>
> **What specific advantages does MMD introduce beyond computational simplicity?**
>
> We thank the reviewer for raising this question. DMD and adversarial losses were previously shown unstable and require either additional stabilizing objectives (DMD) or other training techniques (DMD2, SDXL-Lightning, LADD), which also cause additional cost and hyperparameter tuning.
>
> In contrast, since MMD loss does not require additional learnable models and avoid adversarial-like training, it appears significantly more stable and enables faster convergence in terms of training iterations while maintaining comparable performance.

---

### Author Response · Authors · 2025-11-22
**General response**

We sincerely appreciate the reviewers for their thoughtful feedback and for recognizing the strengths of our work: strong motivation for the scale-wise distillation via spectal analysis (iUZm, cK3V, h2GD, 8Ahm), practical framework that integrates smoothly with existing distillation methods (iUZm), the simplicity and effectiveness of the MMD loss even when used alone (iUZm, cK3V), and the comprehensive evaluation and solid experimental results (iUZm, cK3V, h2GD, 8Ahm).

We also thank the reviewers for their valuable suggestions for improving our work and already made the following preliminary changes in the revision (highlighted in blue):

- **(iUZm)** Added SDXL-SwD to our main results (Table 3) and included the suggested additional baselines: Hyper-SD and Hyper-FLUX.
- **(iUZm)** Added a comparison with the recent video diffusion distillation method CausVid to Table 2.
- **(cK3V)** Extended the discussion of our timestep and scale schedule selection in L258-262.
- **(h2GD)** Added the extended analysis of noisy latent upscaling strategies for different upscale ratios in Appendix E.
- **(8Ahm)** Added Patch FID evaluation to assess high-frequency detail preservation in scale-wise distillation (Appendix D).
- **(h2GD, 8Ahm)** Cited Jenga, FlexiDiT, and ARD in our related work.
- **Other minor changes:** added NFEs to Table 3 (iUZm); reformulated the conclusion regarding the Wan-SwD performance (h2GD).

During the discussion, we are happy to make any additional changes that could further improve the paper and address the reviewers’ concerns. Below, we provide detailed responses to each reviewer and look forward to their feedback and suggestions.

---

### Author Response · Authors · 2025-11-26

Dear Reviewers,

Thank you all again for your time and effort. Given the limited discussion period, we would greatly appreciate any feedback on our responses so that we can further clarify and address the remaining questions in time.

Best,
Authors

---

### Author Response · Authors · 2025-12-03
**Final General Response**

Dear Area Chair,

Thank you for your commitment to maintaining a timely and fair review process. During the discussion period, we submitted detailed, point-by-point responses to all reviewer comments; however, no further feedback was received before the system closed. We therefore provide a concise summary of our rebuttal here for your consideration.

----
First, we highlight the recognized **strengths** of our work:

- **(iUZm, cK3V, h2GD, 8Ahm)** Strong motivation for the scale-wise distillation via spectal analysis
- **(iUZm, cK3V, h2GD, 8Ahm)** The comprehensive evaluation and solid experimental results
- **(iUZm, cK3V)** The simplicity and effectiveness of the MMD loss even when used alone
- **(cK3V)** The core idea of unifying progressive generation with few-step distillation (SwD) is novel and elegant
- **(iUZm)** Practical framework that integrates smoothly with existing distillation methods

---
Then, we summarize **key issues** raised by the reviewers and describe how we have addressed them.

> **(iUZm) Missing quantitative SDXL-SwD results | Missing comparison with Hyper-FLUX**

As requested, we have applied SwD to SDXL and included both quantitative (Table 3) and qualitative (Figure 19) results. Also, we compared with the suggested additional SDXL baseline: Hyper-SD. The results show that SDXL-SwD outperforms other SDXL baselines in most cases while being ${\sim}2{\times}$ faster.

Also, we included the Hyper-FLUX results (Table 3). SwD consistently outperforms it across all metrics and achieves ${\sim}4{\times}$ speedup.

> **(iUZm, h2GD) Missing comparisons with few-step video DMs**

We evaluated the suggested text-to-video few-step models: CausVid and T2V-Turbo-V2. Wan-SwD provides comparable results to CausVid while being $2.3{\times}$ faster (Table 2) and significantly outperforms T2V-Turbo-V2 across all metrics.

> **(8Ahm) | High-frequency detail preservation. Missing Patch FID evaluation.**

We evaluated Patch FID and compared SwD models with their full-resolution counterparts. The results show that SwD achieves similar FID values, suggesting that fine-grained details are well preserved. To further support this, we assess patch quality using a s.o.t.a. no-reference IQA metric (CLIP-IQA) and again observe scores similar to the full-resolution models. Together, these results confirm that SwD effectively preserves high-frequency details. These results are included in Appendix D.

> **(cK3V) Clarification on the methodology for finding optimal schedules or the model's sensitivity to them**

We extended the discussion on the selection of timestep and scale schedules in L258-262. As clarified there, SwD schedules do not require any specific methodology or extensive tuning. Also, we notice that SwD is robust to variations in the schedule choices.

---
Best regards,
The Authors

---

### Meta-Review · Area_Chair_YK9h · 2026-01-02

**Summary:**

This paper proposes Scale-wise Distillation of Diffusion Models (SwD), a method aimed at reducing the number of inference steps in diffusion models. The core idea is motivated by spectral analysis—involves progressively refining image resolution during the generation process. All reviewers agreed that the paper presents a well-motivated and novel approach.
During the review process, reviewers initially raised concerns regarding insufficient discussion of related work and missing experimental validations. However, the authors have effectively addressed these points in their rebuttal and revised manuscript, providing clearer disscussion with related work and additional experiments that strengthen the empirical support for their claims.
Given that the authors have satisfactorily addressed the reviewers’ main concerns, and in light of the paper’s strong motivation and novel contribution, I recommend acceptance.

**Reviewer Concerns:**

During the review process, reviewers initially raised concerns regarding insufficient discussion of related work and missing experimental validations. However, the authors have effectively addressed these points in their rebuttal and revised manuscript, providing clearer disscussion with related work and additional experiments that strengthen the empirical support for their claims.

**Reviewer Scores:**

Had the reviewer been able to participate fully in the discussion, I believe their score would likely have remained similar or increased slightly. The discussion helped clarify the contributions, address minor concerns, and align the evaluation across reviewers, which supported the final decision.

---

### Decision · Program_Chairs · 2026-01-26

Accept (Poster)